# Mechanoelectronic stimulation of autologous extracellular vesicle biosynthesis implant for gut microbiota modulation

Shuangshuang Wan[1], Kepeng Wang[1], Peihong Huang[1], Xian Guo[1], Wurui Liu[1], Yaocheng Li[1], Jingjing Zhang[1], Zhiyang Li[2], Jiacheng Song[3], Wenjing Yang[1], Xianzheng Zhang ✪ [4], Xianguang Ding ✪ [1] ✉, David Tai Leong ✪ [5] ✉ & Lianhui Wang ✪ [1] ✉

Pathogenic gut microbiota is responsible for a few debilitating gastrointestinal diseases. While the host immune cells do produce extracellular vesicles to counteract some deleterious effects of the microbiota, the extracellular vesicles are of insufficient doses and at unreliable exposure times. Here we use mechanical stimulation of hydrogel-embedded macrophage in a bioelectronic controller that on demand boost production of up to 20 times of therapeutic extracellular vesicles to ameliorate the microbes' deleterious effects in vivo. Our miniaturized wireless bioelectronic system termed inducible mechanical activation for in-situ and sustainable generating extracellular vesicles (iMAS-SAGE), leverages on wireless electronics and responsive hydrogel to impose mechanical forces on macrophages to produce extracellular vesicles that rectify gut microbiome dysbiosis and ameliorate colitis. This in vivo controllable extracellular vesicles-produced system holds promise as platform to treat various other diseases.

To allow for proper functioning of the gut, the interactions between the gut cells and the luminal gut microbial communities need to be kept in healthy dynamic homeostasis. Thus it is not surprising that any dysfunctional perturbations to this homeostasis can initiate and precipitate detrimental serious debilitating diseases like inflammatory bowel disease (IBD), neurologic disorders, cardiovascular disease, and even cancer[1–5]. One of the key players is resident macrophages that molecularly interact with these gut microbes. Thus, modulating the gut microbiota itself presents a promising therapeutic opportunity to help treat the many associated diseases[6–8]. Current clinically available microbiota modulation using allogeneic fecal microbiota transplantation (FMT), however, suffers from low-quality control and sustainability

issues for any regular patient recipient dosing. A safety warning on FMT has been raised recently by Food and Drug Administration on inadvertently transferring allogenic antibiotic-resistant microorganisms through FMT[9,10]. On the other hand, conventional gut microbiota modulation approaches, such as dietary probiotics interventions, still face efficacy limitations as they are either ill-defined or lack selectivity to gut site and thereby usually require chronic administration with high dosages to derive any sustainable benefits[11,12]. Consequently, there is an intense interest in developing biosafety microbiota modulation strategies with desirable efficiency.

Host cell-derived extracellular vesicles (EVs) carrying abundant nucleic acids, proteins, and metabolites have been worked as powerful

[1]State Key Laboratory of Organic Electronics and Information Displays & Jiangsu Key Laboratory for Biosensors, Institute of Advanced Materials (IAM), Nanjing University of Posts and Telecommunications, 210023 Nanjing, China. [2]Department of Clinical Laboratory Medicine, Nanjing Drum Tower Hospital, Nanjing University, 210008 Nanjing, China. [3]Department of Radiology, The First Affiliated Hospital of Nanjing Medical University, 210023 Nanjing, China. [4]Key Laboratory of Biomedical Polymers of Ministry of Education & Department of Chemistry, Wuhan University, 430072 Wuhan, China. [5]Department of Chemical and Biomolecular Engineering, National University of Singapore, Singapore 117585, Singapore. ✉e-mail: iamxgding@njupt.edu.cn; cheltwd@nus.edu.sg; iamlhwang@njupt.edu.cn

cellular messengers to facilitate intercellular communications and regulation[13,14]. Recent studies on gut microbiota demonstrated that EVs could serve as long distance non-contact entities to bridge the communication between host cells and microbiome in vivo[15,16]. Host cells have been found to shape microbiota and contribute to improving disordered microbiomes via miRNA-enriched EVs through directly interacting and regulating the transcription and growth of bacteria[16,17]. While EVs are not whole cells, they do partially recapitulate their source cells' bioactivities and nano sizes allowing them to be highly bioavailable and tissue permeability[18]. Within the context of the diseased gut, there is some therapeutic exploitable EVs-mediated cross-communications from intestinal macrophages to the pathological microbiota through EVs. To scale these autologous EVs to reach therapeutic meaningful numbers is a big challenge. Customizable dosing timing is close to impossible as there is no drug induction of specifically targeting intestinal macrophages to produce the necessary EV load. Even if we can obtain sufficient autologous EVs, delivering them to the gut microbiota especially in the small and large intestinal segments is the next big engineering problem. The natural ingestion route will necessitate subjecting the EVs to the degradative acidic gastric component. The anal route would be especially uncomfortable for a regular dosing.

In this work, towards the end of improving the condition of the many patients suffering from IBD, we propose engineering a mesenteric-implanted autologous macrophage bioreactor capable of on-demand in situ production of therapeutic EV that directly provides relief from gut microbiota-induced IBD. The implant is strategically placed at the mesenteric tissues. The mesentery is a dense network of blood vessels and lymph vessels supplying the entire length of the small and large intestines without the need to precisely pin-point

which segments of the entire intestinal tract are affected. Considering that the human intestines are 7.5 m long, it would also exclude the need to subject the human IBD sufferer from painful enteroscopy. The mesenteric implant site also bypasses the EV-degradative unfriendly acidic gastric compartment. We show that mechanical stimulation of macrophages embedded in hydrogel is able to accelerate their production of EV on-demand with our Bluetooth-enabled wireless millimeters actuator. We show that we could tune the EV production with sufficient refinement in the mouse IBD model with significant improvements over control groups. The on-demand therapeutic EV production rate in our system is more than 2000% higher than the natural rate of EV production. The remote wireless control allows for greater portability and greatly lowers the chances for infections arising from open wounds due to external leads. we believe that **i**nducible **M**echanical **A**ctivation for in-**s**itu and **s**ustainable **g**enerating **EV**s (iMASSAGE) would represent the next generation of implant devices that can produce in situ biologics on demand.

## Results

### iMASSAGE technology

During the process of cell signaling in vivo, EVs carrying abundant cellular bioactive substances work as robust mediators to actively facilitate intercellular communication and cross-species talk that influences colonized microbe through direct or indirect interaction[15,17,19] (Fig. 1a). Multimodal characterizations of vesicles derived from host cells confirmed the capability of microbiota to internalize cellular EVs and demonstrated the presence of vesicle-mediated talk cross-species[16,20] (Supplementary Fig. 1). We thus reasoned that host EVs could serve as an inherently safe approach of efficient colonized microbiota modulation in a natural autologous manner and developed iMASSAGE platform

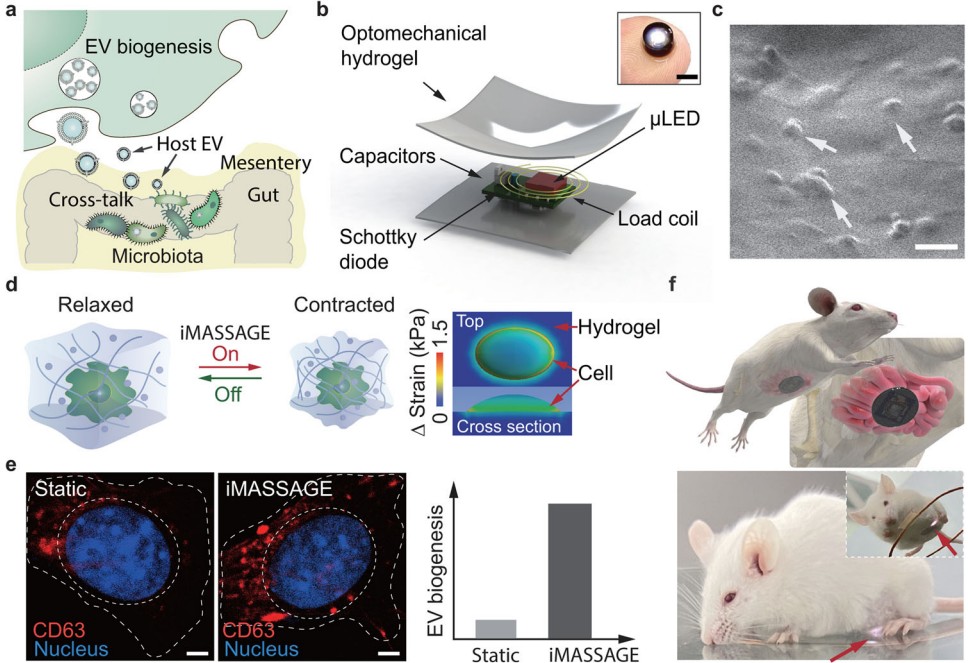

**Fig. 1 | iMASSAGE for in vivo EV production and in situ microbiota modulation. a** Schematics of the biogenesis of EVs and host EVs bridged cross-species communication between host cell and microbiome in vivo. EVs carrying abundant cellular bioactive substances modulating microbiota. Photo credit: Xianguang Ding. **b** Schematic illustration of the implantable iMASSAGE device. Insert shows the iMASSAGE device size. Scale bar, 3 mm. Photo credit: Xianguang Ding. **c** Scanning electron microscopy (SEM) image of cells resided in iMASSAGE device. The white arrows indicate the loaded cells on hydrogel. Scale bar, 20 µm. **d** Left, schematic of embedded cells under iMASSAGE treatment, in relaxed and contracted states. Photo credit: Kepeng Wang. Right, computed distributions of strain

on embedded cells at the top and sectional view after iMASSAGE, assuming an ellipsoidal shape cell adheres on the hydrogel surface. **e** Left, representative fluorescence images of cell immunostained for CD63 after normally cultured (static) and iMASSAGE treated. Scale bar, 2 µm. Right, compared to commonly used static cell culture conditions, iMASSAGE treatment promotes EV biogenesis. **f** Top, schematic illustrations showing the placement of iMASSAGE on the mesentery. Photo credit: JingJing Zhang. Down, a representative image of a freely behaving mice implanted with iMASSAGE device. The white light emitted on the implantation verified function (indicated by a red arrow).

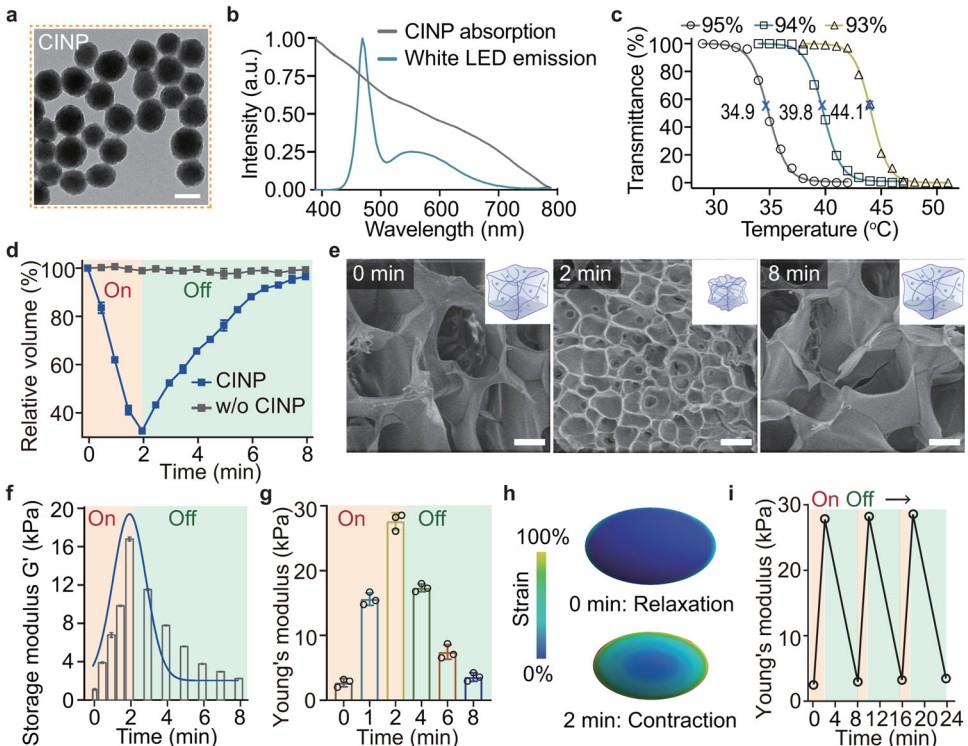

**Fig. 2 | Design and characterization of iMASSAGE technology. a** Representative TEM image of photothermal converter cuttlefish ink nanoparticles (CINPs) for wireless signal conversion. Scale bar, 100 nm. **b** The absorption spectra of CINPs and the emission spectra of white µLED. Adequate spectral overlap brought about subsequent photothermal conversion. **c** Transmittance of the hydrogels with different ratios of NIPAAm (95%, 94%, and 93%) over temperature. The LCST of hydrogels was determined by the temperature at 50% transmittance. **d** Volumetric change of hydrogel-encapsulated iMASSAGE device under wireless-powered stimulation (1 W, with or without CINPs, $n = 3$ independent experiments).

**e** Representative SEM images of the internal aperture of iMASSAGE device in its original state (0 min), the wireless-induced contraction state (2 min) and relaxation state (8 min) observed by SEM. Scale bar: 50 µm. **f** Storage modulus of iMASSAGE device over time (1 W, $n = 3$ independent experiments). **g** Young's modulus by calculating the slope of stress-strain curve of device at 10% strain ($n = 3$ independent experiments). **h** Strain distributions of embedded cell on contractable hydrogel at different time points under stimulation (Top view). **i** The Young's modulus changes of iMASSAGE system as a function of wireless on-off stimulation. All data are presented as mean ± s.d. Source data are provided as a Source Data file.

to manipulate host cells with desired functions to produce abundant bioactive EVs for molecularly modulating gut microbiota (Fig. 1a).

Established on recent findings of mechanical force-engaged EV biogenesis[21,22], the iMASSAGE leverages wireless signal-activated periodic mechanical force posing on embedded cells to modulate the biosynthesis of vesicles and promote EV production. The iMASSAGE device consists of a wirelessly powering µLED that extracts energy from the incident radio-frequency field and optomechanical hydrogel for wireless-regulated mechanical stress generation (Fig. 1b). The wireless signal-responsive hydrogel serves as the wireless-photothermal-mechanical converter and provides overall encapsulation. To support cell residence, arginyl glycyl aspartic acid (RGD) was modified in hydrogel to immobilize cells and minimize cell leakage during operation (Fig. 1c)[23]. The encapsulated system experiences periodic volume contraction and relaxation when wirelessly powered in a radio-frequency field, allowing impulsive stretching force applied on embedded cells (Fig. 1d). Compared to the commonly used static cell culture approach, this iMASSAGE technology leads to enhanced accumulation of intracellular exosomal marker protein and the resultant increment of EV generation (Fig. 1e). All of the manipulations are wireless controllable and programmable by electronic devices to enable in vivo production of therapeutic EVs in an adjustable and repeatable manner for treatment purposes after iMASSAGE system implanted in vivo (Fig. 1f and Supplementary Figs. 2, 3).

In developing the iMASSAGE system, we first designed and synthesized a panel of photothermal-sensitive hydrogels as scaffolds to confer inducible mechanical stress. Cuttlefish ink nanoparticles (CINPs), sourced from natural cuttlefish ink showing superior photothermal

properties and appropriately matched with the optical emission spectrum of µLED, were selected as dopants to facilitate photothermal conversion within the hydrogel and trigger mechanical response (Fig. 2a, b and Supplementary Fig. 4). To meet in vivo-use, the lower critical solution temperature (LCST) of photothermal-sensitive hydrogels was designed between 37 °C and 42 °C. This temperature range ensures that the hydrogel remains stable under physiological conditions (body temperature) and does not harm embedded cells in a short timeframe. With N-isopropylacrylamide (NIPPAm) as the monomer, we optimized the photothermal-sensitive hydrogel through varying precursor ratios (Supplementary Fig. 5a). LCST evaluations demonstrated that hydrogel with an optimized NIPPAm ratio of 94% and CINPs concentration of 1 mg/mL showed the most suitable LCST located at 39.8 °C (Fig. 2c and Supplementary Fig. 5b). Finally, 0.2 mg/mL of RGD was mixed with the optimized hydrogel precursor to induce an in situ radical polymerization on µLED, integrating a wireless activatable device weighing 73 mg with a diameter of 5.2 mm and a height of 3.2 mm.

We first investigated the responsiveness of the device to wireless-powered stimulation. Through controlled wireless stimulating (1 W, 2 min), the hydrogel device experienced drastic shrinkage, with the volume reducing by approximately 70% (Fig. 2d). This deformation can almost be recovered after ceasing the wireless stimulation 6 min later, demonstrating an 8-min of periodical change in volume. Scanning electron micrographs of the freeze-dried device confirmed the reversible contraction of the pores under wireless control (Fig. 2e). In the presence of wireless signals, the pore of the device underwent significant collapse and showed varied size from

50 µm (0 min, original state) to -10 µm (2 min, contracted state), and finally recovered to the original state upon removing wireless stimulation (relaxation state, 8 min).

To gain insight into the mechanical response of the iMASSAGE device and the induced mechanical stress applied to adhesion cells during the cyclic deformation of encapsulated hydrogel, we evaluated their stress-strain metrics. The storage modulus of the hydrogel increased rapidly under 2 min of wireless stimulation, followed by progressively decreasing over time after removing wireless stimulus (Fig. 2f), exhibiting a 6 min of stress relaxation. In a cycle of wireless stimulation, Young's modulus experienced a similar increase from 2.72 kPa to 27.6 kPa, followed by decay close to the original state (Fig. 2g). This varied value was on the order of reported modulus values affecting cell behaviors[22,24]. Finite-element analysis of experiment measurements in the volumetric change of the iMASSAGE device suggested that the induced stress was mainly distributed on cellular membrane attached on hydrogel, assuming an ellipsoidal cell model adheres on the hydrogel surface (Figs. 1d and 2h and Supplementary Fig. 6a). The deformation of the hydrogel and the generated forces led to a complex time-dependent mechanical response of the cell (Supplementary Fig. 6b). Under multiple wireless on-off stimulation, the variations in volume and Young's modulus demonstrated a nearly reversible and repeatable response (Fig. 2i and Supplementary Fig. 7a), suggesting the stability of device's mechanical properties under continuing wireless manipulation. With wireless input power-dependent contractable behaviors, the iMASSAGE features a tunable mechanical profile (Supplementary Fig. 7b–d), suggestive that the mechanical stress of the device imposed on embedded cells could be modulated by varying input power.

The hydrogel encapsulation provided the device with good stability and biocompatibility, which remained functional over 14 days of submersion in PBS buffer without causing apparent toxicity to normal (HUVEC) and immune (RAW264.7) model cells (Supplementary Fig. 8). Moreover, with the 3D macroporous hydrogel structure and RGD functionalization, the device exhibited a high cell loading efficiency of 72% (Supplementary Fig. 9a, b). During a 5-day coculture in the 3D hydrogel scaffolds, embedded cells show minimal growth disadvantages as measured by cellular toxicity assay (Supplementary Fig. 9c, d).

## iMASSAGE promotes EV production

Using the optimized iMASSAGE device, we next evaluated its EV production performance under wireless control. Macrophage cells, the commonly known mechanical-sensitive cells[25] were selected as the model cells to be seeded inside the 3D scaffold of the device. 3D confocal image of the iMASSAGE system revealed the macrophages were dispersed randomly throughout the hydrogel matrix (Supplementary Fig. 10a). The cell embedded device was then exposed to pulsive wireless stimulation without cell damage (Fig. 3a, Supplementary Fig. 10b). To determine EV production yield, culture mediums were collected and the enriched EVs were then counted by standard nanoparticle tracking analysis (NTA) as well as by bicinchoninic acid assay (BCA). We determined that the cells subjected to iMASSAGE could achieve approximately 16.3-fold higher EV production than nontreated control (w/o iMASSAGE, cells resided in iMASSAGE device without wireless stimulation, Fig. 3b and Supplementary Fig. 11a). This enhanced EV generation was further confirmed by specific EV analyses using flow cytometry assay via microbeads functionalized with anti-CD63 antibodies (Fig. 3c and Supplementary Fig. 11b). Transmission electron microscopy revealed the obtained EVs display typical saucer shape with heterogeneity in size and structure (Fig. 3d). Minimal changes were observed in EV diameter and zeta potential across two cell culture conditions (Supplementary Fig. 12a, b). We next evaluated whether iMASSAGE-induced EVs assume similar protein profiles to those of their counterparts (w/o iMASSAGE). Compared with EVs from untreated cells, iMASSAGE-EV not only retained the majority of protein compositions over a broad range of sizes but also demonstrated a significant protein signal enhancement as determined by SDS-PAGE assay (Fig. 3e). Western blotting on the isolated vesicles suggested the presence of typical exosomal markers CD63 and CD9 (Supplementary Fig. 12c). Comparative analyses from proteomics confirmed that two different derived vesicles shared substantial overlay of protein expressions and variations in protein expression were not associated with inflammatory pathways involved in macrophages[26] (Supplementary Fig. 13). In addition, the iMASSAGE-derived EV was also positive for miR-125b and miR-149, known miRNA markers from M2 macrophages (Fig. 3f). The above results indicated that iMASSAGE indeed promoted EVs generation while maintaining biochemical profiles comparable to those of normal-secreted EVs, which supports the use of iMASSAGE device as EV factory for manufacturing bioactive EVs from functional cells.

To verify that the iMASSAGE-induced EVs produce was mainly attributed to the effect of variable mechanics, we incorporated a chemically similar polyacrylamide (PAM) hydrogel as a control device to exclude the potential confounding effects of exogenous radiowave stimulation and temperature changes. This PAM device could be successfully activated and demonstrated similar pore size as the iMASSAGE device (Supplementary Fig. 14a, b). However, mechanical properties of PAM device including volume, Youg's modulus, and pore size, did not change under pulsive wireless stimulation (Supplementary Fig. 14c–e). Utilizing NTA and BCA assay results (Supplementary Figs. 14f, g), we found no significant difference in EV production between the two conditions. This outcome strongly supports the conclusion that the increase in EV production observed with the iMASSAGE device can indeed be attributed to the variable mechanical properties induced by wireless stimulation, rather than to thermal effects or the presence of exogenous radiowaves.

With the tunability of iMASSAGE via wireless, iMASSAGE allowed controllable EV generation with wireless-input power-dependent increasing profiles, as independently validated by NTA and BCA assay (Supplementary Fig. 15), likely as a result of the positive correlation between input power and mechanical stress (Supplementary Fig. 7c, d). A power of 1 W was considered to be the most suitable for subsequent study (Supplementary Fig. 16). Additionally, the wireless stimulation-triggered EV production could also be fine-tuned by stimulation times (Supplementary Fig. 17). Moreover, there was a negligible difference in EVs produced by iMASSAGE device between single- and multiple-pulse wireless stimulation (Supplementary Fig. 18), demonstrating the sustainability of iMASSAGE system in producing EVs. Full activation of the iMASSAGE system could be reached after three times at the input power of 1 W. These parameter sets used for effective EV generation did not induce apparent damage to embedded cells and were selected for the following development of the iMASSAGE platform (Supplementary Fig. 19).

In many disease therapeutics, it may require long-term treatment. We next investigated whether the iMASSAGE system could afford repetitive wireless stimulation to support continuous and reliable EV production during long-term disease treatments (Fig. 3g). Kinetic experiments revealed high viability of embedded cells under multiple days of iMASSAGE treatments with one-day intervals (Fig. 3g and Supplementary Fig. 20). The phenotype of the embedded M2 macrophage cells remained during the continuous stimulation (Supplementary Fig. 21), presumably due to the presence of CINPs with anti-inflammatory polyphenol compounds in the hydrogel[27]. 21 times more EVs were generated even on the fifth day (Fig. 3h and Supplementary Fig. 22), confirming the repeatable working capability of the iMASSAGE device for long-term continuous EV production under the wireless stimulus. This mild and efficient EV production technology can further be expanded to multiple types of cell lines for customized EV generation. We showed a high yield of EVs could be achieved by

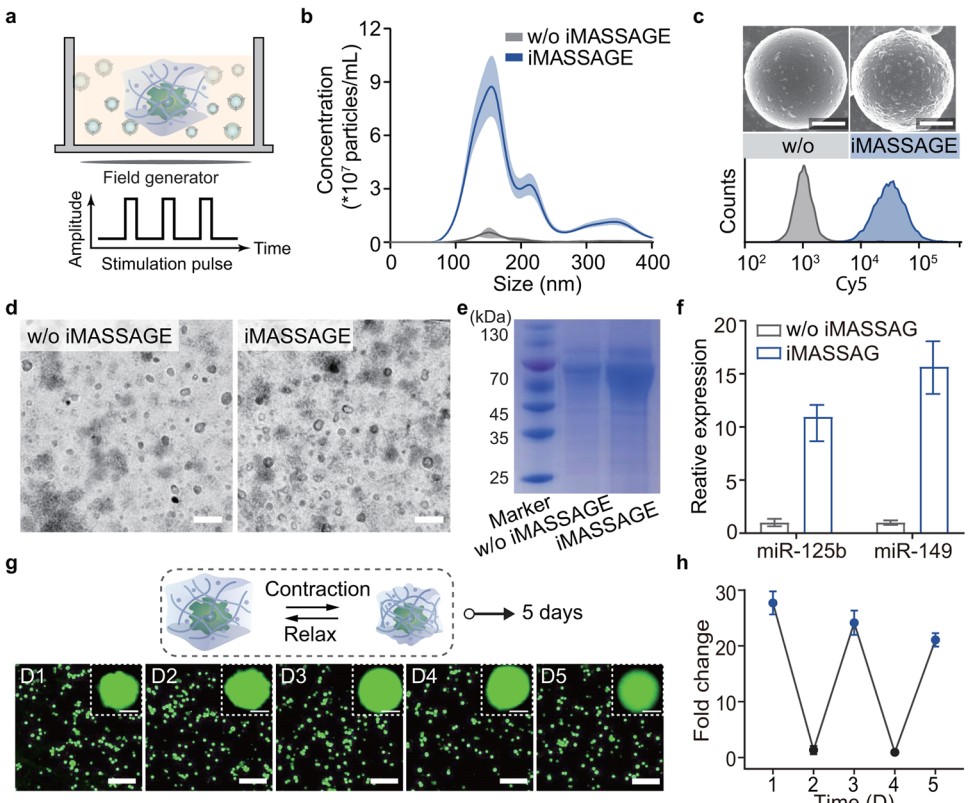

**Fig. 3 | iMASSAGE produces large quantities of therapeutic EV. a** Schematic diagram of iMASSAGE-induced EVs generation under wireless stimulation. **b** Concentration of EVs released from normal culture (w/o iMASSAGE) and iMASSAGE treatment (with iMASSAGE, 1 W, 2 min), measured with NTA. Macrophage RAW264.7 were used. **c** Relative EV amounts detected by flow cytometry. Using CD63 expression as an internal measure of total vesicle counts. Upper: representative SEM images of EVs immuno-captured on anti-CD63 labeled microbeads. More EVs were covered on captured beads for iMASSAGE treatment medium. Scale bar: 400 nm. Down: flow cytometry analysis of bead-bound EVs, after labeling with anti-CD9 conjugated dye. **d** Representative TEM images of EVs collected from normal culture and iMASSAGE treatment, respectively. Scale bar: 200 nm. **e** SDS-PAGE protein analysis of EV collected from different culture conditions. **f** The abundance of miRNA-125b and miR-149 present in normal culture (w/o iMASSAGE) and iMASSAGE treatment-derived EVs, separately. All data were made relative to that of w/o iMASSAGE ($n = 3$ independent experiments). **g** Viability and morphology (upper right) of embedded macrophage cells under multiple days of iMASSAGE treatments with one-day intervals (Calcein AM stained). Scale bar: 10 μm (up) and 50 μm (down). **h** Relative concentrations of EVs produced at different days ($n = 3$ independent experiments). iMASSAGE stimulation (3 T and 1 W, blue dot) was employed on days 1, 3, and 5, without stimulation on days 2 and 5 (black dot). All data were normalized against normally secreted EVs (without stimulation group), obtained without iMASSAGE treatment. All data are presented as mean ± s.d. Source data are provided as a Source Data file.

applying iMASSAGE technology to other mechanosensitive cell lines (Mesenchymal stem cells, skeletal muscle cells MCF-7 breast cancer cells and M1 macrophages, Supplementary Fig. 23). Besides, iMASSAGE system established by other mechanical variable hydrogels, such as 4,4′-di(acrylamido) azobenzene cross-linked polyacrylamide (AZO-PA) hydrogel, had also been verified to accelerate EV production (Supplementary Fig. 24). These suggested the generality of the iMASSAGE technology for promoting the production of EVs.

**Mechanisms of iMASSAGE-regulated EV biosynthesis**
Yes-associated protein (YAP) has been known as a typical cell mechanotransducer, playing a critical role in cell behaviors driven by mechanics[28,29]. To explore the possibility of YAP-driven EV secretion underlying our iMASSAGE technology, we investigated the intracellular YAP expression under iMASSAGE treatment and the corresponding EV secretion. As shown in Fig. 4a, iMASSAGE treatment led to significantly improved endonucleus YAP signal. Consistently, iMASSAGE treatment group showed a higher EVs yield ($8.57 \times 10^7$ particles/mL), more than 10 times than other groups (Fig. 4b). This increased EV generation could be considerably inhibited by the co-incubation of verteporfin (VP, YAP inhibitor). Such characteristic change of EV generation suggested the key role of iMASSAGE-upregulated YAP in the downstream of EV production. To elaborate on the possible pathway

of iMASSAGE-promoted EV secretion via YAP, we further investigated the intracellular multivesicular body (MVB) after iMASSAGE treatment. As the precursor of EV biogenesis, we found the presence of intracellular MVB was in accordance with previous reports that was almost empty and less abundant in the cytosol of standard cultured cells[30] (Fig. 4c). By contrast, cells treated with iMASSAGE generate more MVBs enrichment of intraluminal vesicles. The levels of proteins involved in EV biogenesis were further examined. Comparative analyses confirmed that YAP level with iMASSAGE treatment was significantly higher than untreated group (Supplementary Fig. 25). Notably when compared to untreated cells, the iMASSAGE-treated cells showed an elevated expression level of F-actin, the assembly of which was known to affect YAP nuclear localization[31]. As an important component of the endosome sorting complexes required for transport (ESCRT) machinery for MVBs formation[32], VPS4 proteins had also been up-regulated by iMASSAGE technology. RNA sequencing of embedded cells suggested that iMASSAGE-treated M2 macrophages exhibited significant gene expression differences compared to control groups (Fig. 4d). Of all 357 differentially expressed genes (DEGs) identified (fold change ≥ 2 and $P < 0.05$), 254 were upregulated and 103 were downregulated. It was worth noting that mechanosensing genes (YAP1) and genes related to EV biosynthesis (VSP4a, VSP4b, SNF8, and TSG101) were significantly upregulated. Besides, the Gene Ontology

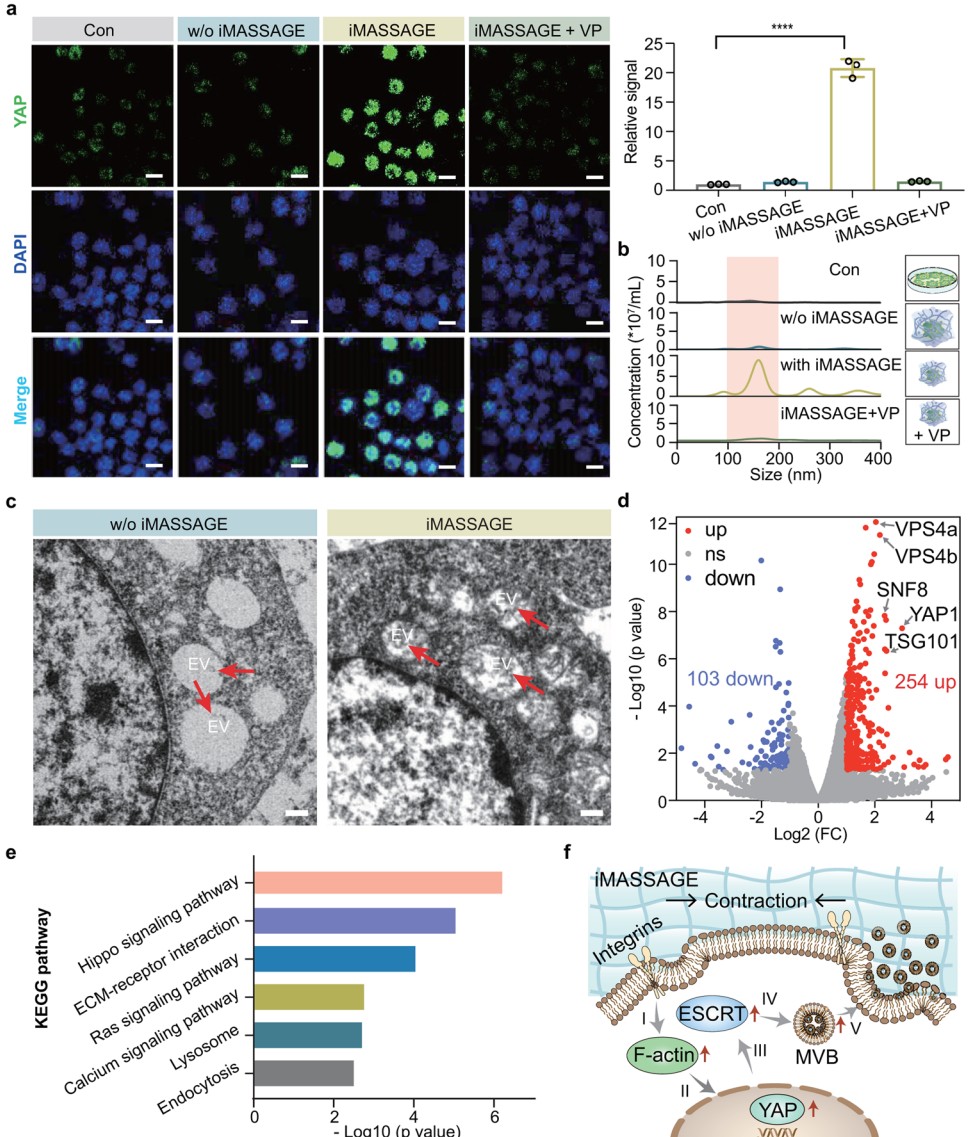

**Fig. 4 | Mechanisms of iMASSAGE-mediated EV biosynthesis. a** Left, YAP expression in the nucleus of cells residing in the iMASSAGE system after different treatments. Anti-YAP (green) and DAPI (blue) were used for staining. VP, YAP inhibitor. Scale bar: 10 μm. Right, intracellular YAP signal after different treatments ($n = 3$ independent experiments). **b** Quantification of EV concentrations in the culture medium after different treatments 24 h later. The values in the graph indicate the concentration at the NTA curve. **c** TEM images of cells resided in the system with or without iMASSAGE stimulus. The red arrows indicate intraluminal vesicles containing EVs. Scale bar: 500 nm. **d** Volcano plot displaying differences in RNA gene expression in M2 macrophages by comparing iMASSAGE and control groups. **e** KEGG pathway analysis by comparing iMASSAGE-promoted EV release associated with different genes. **f** Proposed mechanism for iMASSAGE-triggered EV generation. **I** Activation of F-actin by iMASSAGE. **II** Upregulation of intranuclear YAP. **III** Upregulation of ESCRT. **IV** Accelerated MVB formation. **V** EV secretion. The significant differences were calculated based on two-tailed Student's $t$ test. \*\*\*\*$p < 0.0001$. All data are presented as mean ± s.d. Source data are provided as a Source Data file.

(GO) analysis of the top 20 shows that biological processes and intracellular components involved in cellular mechanosensing and EV release were successfully activated, including Hippo signal and ESCRT complex (Supplementary Fig. 26). Furthermore, six major signaling pathways such as Hippo signaling pathway, ECM-receptor, and Ras signaling pathway, were altered in Kyoto Encyclopedia of Genes and Genomes (KEGG) pathway analysis (Fig. 4e). These demonstrate that iMASSAGE can trigger intracellular signaling alterations to regulate cellular features toward EV biosynthesis.

Together, these data revealed the key role of YAP in iMASSAGE-stimulated EV biosynthesis. Specially, we propose in our iMASSAGE model that cells residing in device were subjected to mechanical force from iMASSAGE technology. This activates the intracellular F-actin following the sense of membrane protein, such as integrin, and results

in an upregulation of the expression of YAP in nucleus, which further facilitates the formation of MVB aided by ESCRT system and contributes to productive EV generation (Fig. 4f).

## iMASSAGE produces EVs for microbial modulation

Motivated by EV-mediated cross-species communication between host and microbe in vivo[16,17], we reasoned that EVs carrying cellular bioactive contents could serve as efficient shuttles to influence microbe through direct or indirect interaction. Manipulating cells with desired functions to produce abundant bioactive EVs, for example, M2 macrophage-generated EVs, could thus present an approach to molecularly modulate gut microbiota. M2 macrophage with rectifying capability plays an intrinsic role in reshaping dysregulated gut microbiota and has been demonstrated to be essential in maintaining a balanced microbiota[33].

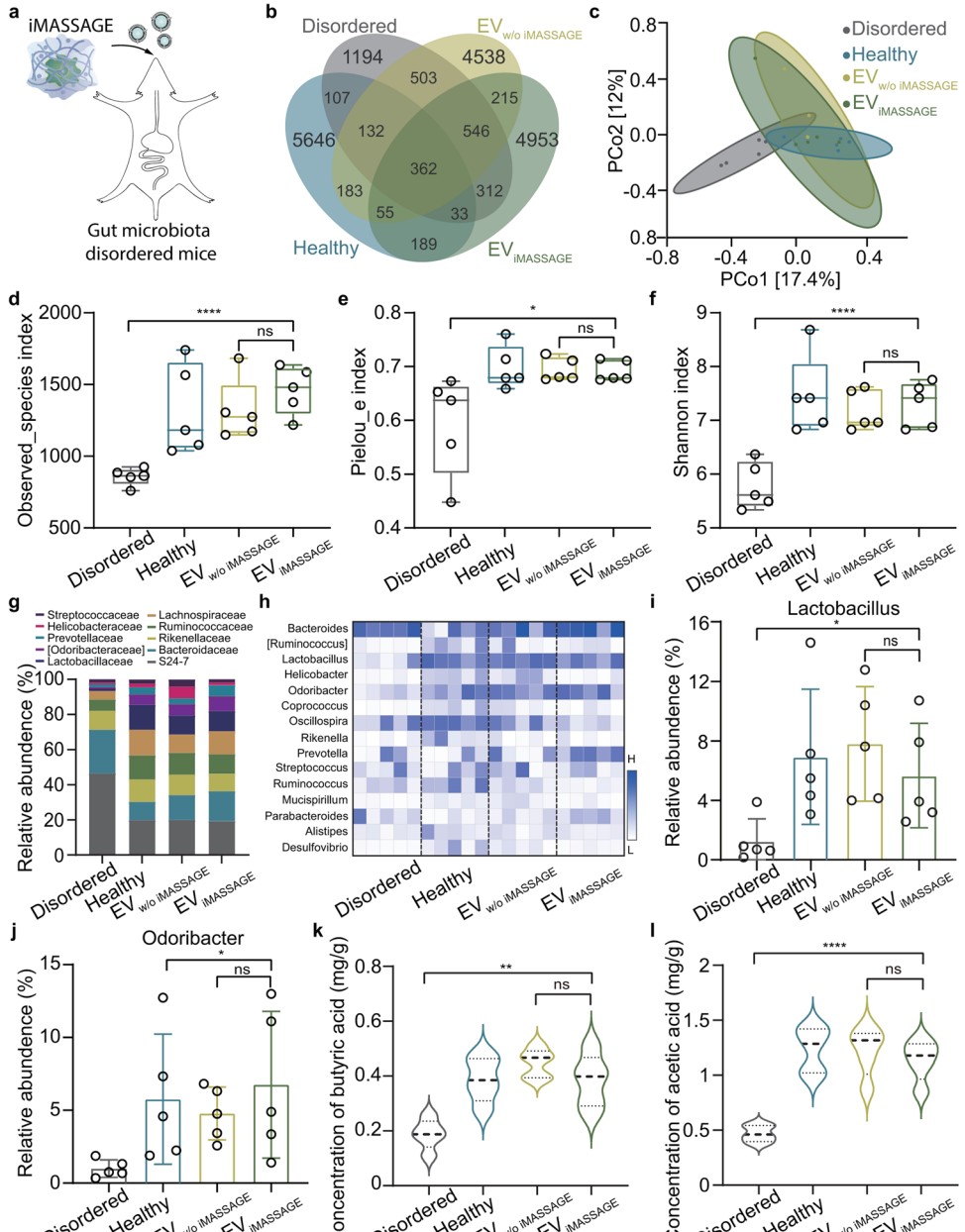

**Fig. 5 | iMASSAGE generated EVs for microbiota modulation. a** Mice with disturbed gut microbiome were orally administered with equal amounts of EVs ($1.0 \times 10^{10}$ particles/g, M2-typed macrophage originated) obtained from iMASSAGE and commonly cultured strategy, respectively. The feces were collected for microbiota analysis by 16s RNA sequencing on day 7. **b** Venn diagram of all bacterial strains identified from differently treated mouse feces. **c** Principal coordinate analysis (PCoA) of the Bray-Curtis distance based on OTUs after different treatments. Observed_species index (**d**), Pielou_e index (**e**), Shannon index (**f**) of gut microbiota from mice feces for assessing the α diversity of the gut microbiome ($n = 5$ mice per group). Minima: Lower limit of the whisker; Maxima: Upper limit of the whisker; Center: Median line inside the box; The upper and lower box bounds represent the 25% and 75% percentile of data. **g** The relative abundance of the top 10 microorganisms at the family level. **h** Heatmap of the abundance of the top 15 microorganisms at the genus level. The relative abundance of (**i**) *Lactobacillus* (**j**) *Odoribacter* ($n = 5$ mice per group). The concentration of (**k**) acetic acid and (**l**) butyric acid in the feces on the seventh day detected by GC-MS ($n = 5$ mice per group). Data present mean ± s.d. The significant differences were calculated based on two-tailed Student's *t* test. \*\*\*\**p* < 0.0001, \*\**p* < 0.01, \**p* < 0.05, ns refers to no significant. All data are presented as mean ± s.d. Source data are provided as a Source Data file.

After verifying that EVs could successfully enter the intestinal lumen through oral administration (Supplementary Fig. 27), we then leveraged M2 macrophage-embedded iMASSAGE system to generate EV (EV_iMASSAGE) and evaluated their performance on microbial modulation (Fig. 5a and Supplementary Fig. 28a). As a comparison, EVs generated through the commonly static culture strategy of M2 macrophages (EV_w/o iMASSAGE) were also collected and used as control. With equal EV administration amounts, we found EV_w/o iMASSAGE and EV_iMASSAGE treatments significantly increased the total number of unique operational taxonomic units (OTUs) when compared to the untreated gut disordered group (Fig. 5b and Supplementary Fig. 28b). Both EV_w/o iMASSAGE and EV_iMASSAGE restored microbes close to a level similar to that of healthy control. Using the principal coordinate analysis (PCoA), we found that the gut microbial profiles of EV_iMASSAGE and EV_w/o iMASSAGE treatments exhibited distinct clusters from disordered group (P(EV_iMASSAGE VS Disordered)= 0.007, P (EV_w/o iMASSAGE VS Disordered) = 0.006, Fig. 5c), but overlapped well with those of healthy control. This

illustrated the positive regulatory effect of macrophage-derived EVs on the composition of microbiota communities. More importantly, compared with the negative control group (gut disordered), $EV_{iMASSAGE}$ and $EV_{w/o\ iMASSAGE}$-treated mice had a significant increase in three microbial α diversity indexes (Fig. 5d–f). Besides, $EV_{w/o\ iMASSAGE}$ and $EV_{iMASSAGE}$ groups shared similar microbial proportions to healthy group in the top 10 dominant strains (Fig. 5g), which was further manifested by heatmap analysis of flora fractions at genus level (Fig. 5h). These observed similarities in gut microbial shaping behaviors of M2 macrophage-EVs obtained from different access approaches may be attributed to the presence of cell-derived bioactive contents, which influence microbiotas through direct or indirect interaction[16,17,34,35].

We also measured the abundance of beneficial bacteria in various treatments. Across different treated mice, $EV_{iMASSAGE}$ and $EV_{w/o\ iMASSAGE}$ markedly increased two important beneficial bacteria: *Lactobacillus* and *Odoribacter*, which were known to reduce inflammation and repair the intestinal barrier[8,36] (Fig. 5i, j). In addition, we found $EV_{iMASSAGE}$ treatment significantly elevated the content of acetic acid and butyric acid (Fig. 5k, l), the critical microbial metabolites in maintaining the normal function of intestine[37,38]. Such increased trends were consistent with $EV_{w/o\ iMASSAGE}$. No statistically significant difference was found between $EV_{w/o\ iMASSAGE}$ and $EV_{iMASSAGE}$ treatment. These agreements not only validated that EVs were good cellular surrogates for gut microbiota modulation but also indicated similar functions between $EV_{iMASSAGE}$ and $EV_{w/o\ iMASSAGE}$, thereby potentiating the use of iMASSAGE platform to generate rich EVs as cellular surrogates in disease treatment.

## Implanted iMASSAGE restores microbial homeostasis for colitis treatment

To evaluate the clinical applications of iMASSAGE platform, we conducted a feasibility study using IBD as the therapeutic model. IBD is a common and chronic heterogeneous disease with complex conditions at the clinical that calls for individualized treatment. Microbiota dysbiosis has been identified as a risk factor in the development of IBD and so intestinal microbiota is an attractive target for IBD treatment[39]. We aimed at addressing (1) whether iMASSAGE system could work in vivo to produce rich EV, (2) the efficiency of implanted iMASSAGE for microbiota modulation and (3) the potency of iMASSAGE in treating IBD under wireless signal control (Fig. 6a).

We surgically implanted the device at the mesentery near the colonic site (Supplementary Fig. 29). The implanted iMASSAGE system could be effectively activated in vivo through wireless manipulation, as demonstrated by the transdermal light spot (Fig. 1f and Supplementary Fig. 29). With the optimized input as in vitro experiments (1 W, three times), around 2.3 °C of temperature increment was observed (Supplementary Fig. 30). This increased temperature remained within the range that is considered physiologically tolerable for the tissue involved[40,41]. In addition, we observed that the iMASSAGE device was retained at the implanted site and appeared to be biocompatible a week later (Fig. 6b). Implanted cell-free iMASSAGE device in vivo demonstrated that the device itself under plusive wireless stimulation had little influence on the secretion of EVs in surrounding tissue cells (Supplementary Fig. 31). Then, we constructed a macrophage cell line that stably express CD63-GFP protein (Supplementary Fig. 32a, b), which was able to secrete EVs containing CD63-GFP protein (Supplementary Fig. 32c). The stably transduced macrophages were loaded into device (Supplementary Fig. 32d) to specifically track the EVs produced by the embedded macrophages. We collected colonic tissues near the implant and compared the levels of generated EVs diffused in intestinal tissues using flow cytometry and immunostaining. The results showed that iMASSAGE treatment led to a higher level of vesicular markers in colonic tissues after 72 h (Fig. 6c and Supplementary Fig. 33), while those

without wireless activation showed low vesicular markers, suggesting the successful activation of implanted iMASSAGE on accelerating EV generation and release in vivo. In addition, we observed the CD63-GFP fluorescence of EVs in the intestinal lumen (Supplementary Fig. 34a). The flow cytometry analysis of the intestinal lumen contents further demonstrated a higher accumulation of CD63-GFP-EVs in the treated groups compared to the group without iMASSAGE treatment (Supplementary Fig. 34b).

We next evaluated if iMASSAGE could shape the homeostasis of colonic flora in vivo. With the implantation of therapeutic M2 macrophage-embedded iMASSAGE system, it was found that the reduced OTUs in colitis mice were significantly elevated after iMASSAGE activation (Supplementary Fig. 35a). We further observed a considerable difference of gut microbial profiles ($P = 0.008$) between Sham and iMASSAGE groups (Supplementary Fig. 35b). A comparative analysis of the microbiota diversity index of colitis mice revealed that iMASSAGE treatment improved the diversity, richness and uniformity of the flora (Supplementary Fig. 35c). Two typical beneficial bacteria (*Lactobacillus* and *Odoribacter*) were observed to be abundantly present as a result of iMASSAGE (Fig. 6d and Supplementary Fig. 35d). In addition, the iMASSAGE-treatment group experienced a significant increase in short-chain fatty acids (acetic acid and butyric acid, Fig. 6e, f). Together, these results suggested that the microbiota of colitis mice was successfully shaped by iMASSAGE system, which supported the use of iMASSAGE technology as an in vivo EV production platform to modulate the intestinal microbe in situ for therapeutic purposes.

We next examined the treatment efficacy of implanted iMASSAGE system for IBD (Supplementary Fig. 36a). We found IBD mice treated with M2 macrophage-embedded iMASSAGE remain maintained body weight above 91.5% and shown a decrease of disease activity index (DAI) with longer colons compared to controls (Fig. 6g–i and Supplementary Fig. 36b, c). Histological analysis confirmed that iMASSAGE therapy effectively reduced the infiltration of inflammatory cells in colonic tissues and repaired DSS-damaged colon structures without causing noticeable side effects to normal organs (Fig. 6j and Supplementary Figs. 37, 38). Moreover, iMASSAGE treatment upregulated anti-inflammatory factor (IL-10) and decreased pro-inflammatory cytokines (IL-6, IL-1β, and TNF-α) in serum (Fig. 6k and Supplementary Fig. 39). Notably, compared to traditional systemical therapy by oral administration of EVs, this iMASSAGE-mediated in situ EV production strategy showed superiority in colitis treatment (Supplementary Fig. 40), which may be attributed to the abundant localized EV generation from implanted iMASSAGE device and their resultant increased bioavailability without dosage depleting in circulatory system.

## Discussion

Host EVs play a critical role in bridging the communication between cells and microbiome in vivo and offer a novel form of natural autologous therapeutics for efficient microbiota modulation and disease intervention. The established iMASSAGE platform opened up the possibilities for rapid in vivo manufacture and release of functional EVs to regulate the microbiome, which not only presents distinct technological advances for EV therapeutics but also provides an in-situ handle, enabling controllable EV biogenesis from determined cells for enhanced understanding of EV-mediated intercellular and cross-species communication within living organisms.

In relation to treatment innovation, the on-demand iMASSAGE system offer adaptability, precision, and control in response to the dynamic nature of diseases. Unlike static hydrogels that cannot adjust to the fluctuating therapeutic needs of progressing diseases, the iMASSAGE system can modulate the release of EVs according to the specific stages of disease or healing. This adaptability ensures that therapy is tailored to the temporal dynamics of the condition,

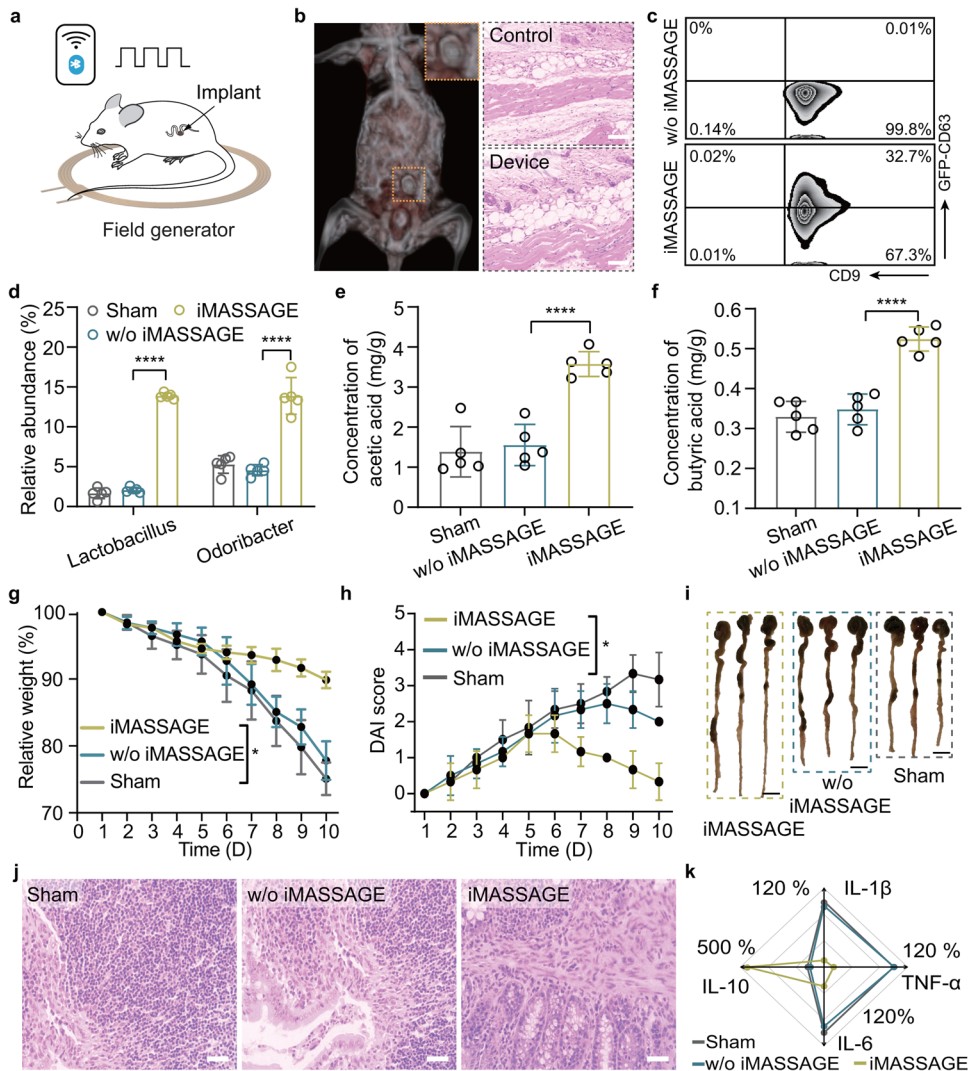

**Fig. 6 | Implanted iMASSAGE restore microbiome homeostasis for IBD treatment. a** Scheme of implanted iMASSAGE producing on-demand EV in treating IBD under wireless signal control. **b** Left: CT images of iMASSAGE device after surgical implantation in mice seven days later. Right: H&E staining of colonic tissue near the retrived device. The implantation shows good tissue biocompatibility compared to tissues without iMASSAGE device. Scale bar: 100 μm. **c** Detection of EVs in colon tissue from device resided the stably transduced macrophages with or without pulsed wireless stimulation by flow cytometry. **d** The relative abundance of *Lactobacillus* and *Odoribacter* after different treatments (*n* = 5 mice per group). The concentration of (**e**) acetic acid and (**f**) butyric acid in the feces of mice detected by GC-MS (*n* = 5 mice per group). Relative weight (**g**) and (**h**) DAI score of mice after different treatments (*n* = 5 mice per group). **i** Images of colon tissues ex vivo from different treated mice on day 10. Scale bar: 1 cm. **j** Representative H&E images of colon tissue. Scale bar: 50 μm. **k** Radar plot of the expression of inflammatory factors in blood after different treatments. The significant differences were calculated based on two-tailed Student's *t* test. ****$p < 0.0001$, *$p < 0.05$, ns refers to no significant. All data are presented as mean ± s.d. Source data are provided as a Source Data file.

providing a more targeted approach that can accommodate phases of exacerbation and remission. Furthermore, the system's capacity for controlled dosing significantly reduces the risk of therapeutic resistance or adverse side effects, which are common challenges with continuous exposure to therapeutic agents. By further integration with external monitoring and control systems, the dynamic nature of the on-demand hydrogel system could allow for real-time adjustments based on physiological feedback, further enhancing the precision and adaptability of the therapy.

For technological advancement, the iMASSAGE system integrates wireless bioelectronics and stretchable hydrogel to program the biosynthesis of EVs in cells and achieve high production yield. With respect to EV production, currently developed large-scale EV manufacturing strategy commonly uses scale-up cell culture routes, such as multilayered culture flasks or bioreactors to expand cell culture capacity[42]; these methods are costly, sophisticated facilities rely on and challenged in cell senescence during expansion. In comparison,

the iMASSAGE strategy improves the yield of EVs with available cells by boosting the productivity of individual seeded cells. In this term, the iMASSAGE strategy can further be integrated into existing expanded cell culture approaches, such as bioreactors, to improve the yield scale of EVs further. With respect to EV therapeutics, the iMASSAGE technology offers a remarkable capability of wireless control and programming, enabling remote control of EV biosynthesis. This feature is attractive for in vivo applications following implantation. Furthermore, the implanted iMASSAGE supports EV production in vivo for directly localized intervention, thus eliminating the requirement of extensive ex vivo cell culture to acquire sufficient quantities of EVs for treatment purposes.

The scientific applications of the established technology are potentially broad. Recent studies have increasingly recognized the interplay between macrophages and the gut microbiota. EVs are known to carry a myriad of bioactive molecules, including proteins, lipids, and nucleic acids, which can significantly impact the recipient

cells and the surrounding microenvironment. For instance, studies by Ji et al. and Liu et al. suggested that host EVs could carry specific miRNA and enable reshape the composition of intestinal gut microbiota[17,43], suggesting a potential regulatory role of host EVs. EV-miRNA from monocytes and macrophages can further alter Intestinal epithelial cells function and intestinal barrier by modulating the activity of inflammatory transcription genes[44]. In our study, we observed an increase in specific bacterial populations in the gut following the administration of M2 macrophage-derived EVs post-antibiotic treatment. This observation aligns with the hypothesis that EVs can promote the growth of gut microbiota. We propose several mechanisms through which this might occur: (a) direct stimulation. EVs may directly stimulate the growth of certain bacterial strains by delivering cellular components or substrates[45]. (b) immune modulation. By modulating the immune response, EVs could create a more favorable gut environment for the growth of specific bacteria while inhibiting others. Emerging researches have expanded the understanding of the interplay between intestinal microbiota and various diseases. Microbiome has also been found in cancerous tissue. With host EV mediated inherently interaction with microbiome and the robust performance of iMASSAGE in producing EV in vivo, we anticipate the technology can be applied to multiple EV therapeutics, such as cancer therapy and diabetic treatment[46,47]. The iMASSAGE technology provides a modular platform readily adapted to various mechanosensitive cells for therapeutic EV manufacturing. Technical improvements, through engineering functional cells with the expression of mechanosensor such as Piezo1 and genetic transducing modules should enhance the perception of mechanics generated by wireless-mediated iMASSAGE and further accelerate the mechano-genetic transduction into cellular activities of EV biogeneration. Further technical advances, such as miniaturization and flexibilization of the bioelectronic device to facilitate the iMASSAGE system adapts to the contour and biointerface of tissues[48–50], could enhance its performance and accelerate clinical validation. Further integration with other biological barrier crossing phenomena like nanomaterials induced endothelial leakiness (NanoEL) may expand the possibilities of these important cell derived nanomaterials' translatability into other diseases of global concern[51–54].

## Methods

### Ethics statement

All research complied with all relevant ethical regulations. The animal experiments were conducted in accordance with the approved protocol of the Committee for Animal Research of Nanjing University of Posts and Telecommunications (No: 202202). BALB/c mice (6–8 weeks, female) were purchased from Beijing Vital River Laboratory Animal Technology Co., Ltd. All mice (5 mice per cage) were housed in a specific pathogen-free environment at a temperature of 18–22 °C and a humidity of 50–60%. All mice were be promptly euthanized using $CO_2$ gas after finishing animal experiment.

### Materials

NIPAAm, Methylene-N, N-bis(acrylamide) (MBA), ammonium peroxodisulfate (APS), acrylamide (AAm) and acrylamide (AM) were purchased from Sigma-Aldrich. Arg-Gly-Asp (RGD) peptide was obtained from MedChemExpress. Calcein-AM, cell Counting Kit-8 (CCK8) and BCA protein assay kits were purchased from Beyotime Biotech Co., Ltd. All Elisa kits were brought from 4 A Biotech Co., Ltd.

### Endocytosis of EVs by microbiota

EVs derived from macrophages (RAW264.7, ATCC: TIB-71) and human intestinal epithelial cells (ATCC: HIEC-6) were collected by gradient centrifugation. Briefly, cell culture supernatants were centrifuged at 2000 g for 10 min, followed by centrifugation at 10,000 g for 30 min. The final EVs were obtained by collecting the pellet after centrifugation (100,000 g, 2 h). DiD-labeled EVs (20 μg/mL, 200 μL) were co-cultured with Calcein-AM-stained gut microbiota (10^7 CFU/mL), including *Staphylococcus aureus* (ATCC 23235, *S. aureus*,) and *Escherichia coli* (ATCC 25922, *E. coli*) for 4 h, respectively. After washing three times wirth PBS (10,000 g, 10 min), the uptake of EVs by microbiota was observed using a laser scanning confocal microscope (Leica TCS SP8, Germany) and analyzed by flow cytometry (BD Accuri C6).

### Extraction of cuttlefish ink nanoparticles (CINPs)

The CINPs were extracted from cuttlefish ink sacs by gradient centrifugation. Briefly, ink sacs were obtained from fresh cuttlefish and dispersed in deionized water (DI water). The solution was centrifuged at low speed (600 g) for 5 min to remove large impurities, followed by high-speed centrifugation at 12,000 g for 15 min to obtain CINPs. The pelleted CINPs were washed by DI water three times and suspended in DI water for use.

### Synthesis of photoresponsive hydrogel

The photoresponsive hydrogels were synthesized by in situ atom-transfer radical polymerization. We first obtained the polymer precursor solution by dissolving 57 mg of N-isopropylacrylamid (NIPAAm), 3 mg of acrylamide (AAm), and 6 mg of methylene-N, N-bis(acrylamide) (MBA), into 1 mL deionized (DI) water. Then, 50 μL of CINPs (20 mg/mL), 100 μL of N, N, N', N'- tetramethylethylenediamine (TEMED), 100 μL of ammonium peroxodisulfate (10 mg/mL) were added into the above precursor solution. The homogeneous mixture was immediately added to the round molds. 10 min later, the pNI-PAAm-co-AAM/CINPs hydrogels were formed and dialyzed several times in DI water to remove the non-polymerized monomers.

### Lower critical solution temperature (LCST) measurement

To achieve phase transition of the pNIPAAm-co-AAM/CINPs hydrogels in vivo (37 °C < LCST < 42 °C), we synthesized three photosensitive hydrogels by changing the NIPAAm ratio (95%, 94% and 93%). The absorbance (450 nm) of the hydrogels in the range of 29 °C–51 °C was recorded using at a heating rate of 1.0 °C/min. The LCST was determined by the temperature corresponding to the fitted curve at 50% absorption.

### Photothermal responsiveness of hydrogel

1 mg/mL CINPs were dispersed in DI water, and irradiated by Xenon lamp (CEL-LB50, China). The temperature of the solution over irradiation time was recorded by a near-infrared thermal imager (FLIR A655sc). The same method was used to detect the temperature change of the hydrogel with or without CINPs over time.

### iMASSAGE device construction

μLED devices were assembled following the procedure reported by John S. Ho and co-workers[55]. The wireless optical transmission device includes a coil for receiving radio frequency energy, a light-emitting diode, capacitors, and Schottky diodes. The LED and the capacitor were connected in series, and the emission was white light with an optical spectrum centered at 460 nm, corresponding to the absorption peak of the photosensitive hydrogel. The wireless powered-μLED was then pre-sealed with optically transparent silicone (polydimethylsiloxane, PDMS) to prevent the leakage of electronic components. Then, the iMASSAGE device was fabricated by encapsulating the μLED inside the thermoresponsive hydrogels. Briefly, 50 μL CINPs (20 mg/mL), 100 μL TEMED, 100 μL of ammonium peroxodisulfate (10 mg/mL) were added into the polymer precursor solution (57 mg NIPAAm, 3 mg AAm, 6 mg MBA and 0.2 mg RGD peptide). After mixing the solution evenly, the mixture was immediately added to the round mold containing the PDMS pre-sealed μLED. The iMASSAGE device was obtained after 10 min of polymerization. After that, the devices were dialyzed several times and frozen at −20 °C for 1 h.

## Polyacrylamide (PAM) device construction

Similar to construction process of the iMASSAGE device, the PAM device was fabricated by encapsulating the μLED into the PAM hydrogels. Briefly, 0.3 g AM, 0.09 mg MBA and 0.2 mg RGD peptide were added into 1 mL DI water and stirred for 30 min. Then, 50 μL CINPs (20 mg/mL), 0.54 μL TEMED, 0.65 mg ammonium peroxodisulfate were added and stirred for 30 min. After 10 min of ultrasound, the mixture above was added to the round mold containing the PDMS pre-sealed μLED and react for 12 h at 30 °C. The PAM device was dialyzed several times and frozen at −20 °C for 1 h.

## Evaluation of the wireless responsiveness of iMASSAGE device

The volume of the iMASSAGE device was measured every 30 s during 8 min on-off wireless stimulation (1 W, 2 min of On and 6 min of Off). At 0 min, 2 min, and 8 min, the macropores of freeze-dried devices were observed by scanning electron microscopy (SEM, S4800). Then, we detected the storage modulus of the device using a rheometer (DHR-1) at pre-set time (0, 1, 2, 4, 6, and 8 min) of a iMASSAGE stimulation (8 min). Besides, the mechanical properties of the device were tested at pre-set time by an electromechanical universal testing machine (CMT6502) in standard stress/strain experiments. And the corresponding Young's modulus was determined from the slope of the first 10% stress-strain curves. To illustrate the reversibility of the device's wireless stimulation, the volume and Young's modulus of the device were measured in the maximum compression and expansion state. Additionally, the temperature of the device over time was recorded by a near-infrared thermal imager after different wireless input powers (0, 0.5, 1, and 2 W) treatment. The Young's modulus of the device at the maximum compression state under different wireless input powers was detected.

## Cell culture

The RAW264.7 murine macrophage cells and human intestinal epithelial cells were purchased from ATCC. RAW264.7 cells were maintained in RPMI 1640 medium (Invitrogen) supplement with 10% (v/v) fetal bovine serum (FBS, Gibico) and 1% penicillin/streptomycin. HUVEC cells were cultured in ECM medium (ScienCell) supplement with 10%(v/v) fetal bovine serum (FBS, Gibico). Mesenchymal stem cells were maintained in Mesenchymal Stem Cell Basal Medium (MSCBM, DAKEWE); Skeletal muscle cells were maintained in Skeletal Muscle Cell Medium (SkMCM, ScienCell) supplement with 10% (v/v) fetal bovine serum (FBS, Gibico) and 1% penicillin/streptomycin; MCF-7 cells were cultured in Dulbecco's modified Eagle medium (DMEM, Invitrogen).

## Cytotoxicity of the device

HUVEC and RAW264.7 as model cells were seeded in 96-well plates with 3000 cells per well. Subsequently, cells were treated with the leaching solution that the device soaked in the medium for 5 days. Cell viability was detected with cell counting kit-8 (CCK-8) according to the manufacturer's instructions. Additionally, the cytotoxicity of the device was also examined with a transwell assay. HUVEC and RAW 264.7 cells were seeded in 24-well plates. The device was added to the upper chamber of the 24-well plate and activated with pulse wireless signal (1 W, 1 T). The cell viability was analyzed according to the manufacturer's instruction of CCK8. M2 macrophages-resided device was culture in medium. At determined days, the culture medium was collected for LDH assay. LDH released was detected with a lactate dehydrogenase assay kit according to the manufacturer's instructions.

## Cells loading in the device

RAW264.7 cells were incubated with IL-4 (20 ng/mL) for 24 h to obtained M2 macrophages. 1 mL of cell medium containing M2 macrophages ($1 \times 10^6$/mL) was then dripped onto the device in a dried state and incubated for 12 h. Then the device was washed with fresh cell medium to remove unloaded cells. The number of cells in the culture medium before and after device immersion was counted using Z2™ Coulter Counter (Invitrogen Countess 3). The loading percentage of cells was calculated by the formula: $(N_{before}-N_{after})/N_{before} \times 100\%$. Besides, the cell-resided device was fixed and lyophilized, followed by observation with SEM.

## Distribution and viability of cell loaded in device

CINPs was labeled with tetracarboxyporphyrin by esterification reaction for device construction. M2 macrophages-resided device were prepared based on the above method. Then, device was stained with 0.02 mM Calcein-AM for 30 min. After washing to remove excess dye, the device was stored in −80 °C to obtained frozen section, following by observation using fluorescence confocal microscopy. Besides, cell-resided device was treated with pulse wireless signal (1 W, 1 T). 12 h later, the device was stained with Calcein-AM and observed by fluorescence confocal microscopy for cell viability assessment.

## Finite element analysis of hydrogel scaffold

Finite element analysis was performed using commercial software ANSYS (ANSYS 2019 R2). An ellipse-shaped cell with a height of 2 μm, length of 15 μm and width of 8 μm on a square hydrogel substrate unit (width: 50 μm, height: 2 μm). We used 14195 mesh elements for modeling of the single cell-attached hydrogel. The cell was modeled as isotropic materials with Poisson's ratio of 0.49 and Young's modulus of 10 kPa. The Poisson's ratio of hydrogel was set as 0.47 and Young's modulus was dependent on the varied temperature determined by fitting with the initial modulus (1071 Pa).

## Flow cytometry assay

RAW264.7 cells were differentiated by treating with 20 ng/mL of interleukin-4 (IL-4) for 24 h to obtain M2 macrophages. Subsequently, 1 mL of cell medium containing M2 macrophages ($1 \times 10^6$/mL) was dripped onto the device in a dried state, followed by incubating for 12 h. After subjecting the M2 macrophage-loaded devices to various treatments, the devices were incubated with trypsin and washed with fresh cell medium to collect cells. The obtained cells were then incubated with anti-CD11b and anti-CD86 antibodies for 20 min at 4 °C and then washed with PBS. Anti-CD206 antibody was incubated with cells for 60 min on ice, following the use of a fixation/permeabilization kit (BioLegend). All samples were analyzed by flow cytometer (BD Accuri C6) and the data were analyzed using the FlowJo v10.8.1.

## EV isolated from device and characterization

The M2 macrophage-resided iMASSAGE device was cultured in EV-free culture medium. After different treatments, the device was cultured at 37 °C for 24 h. Then the medium was collected and the device was washed with fresh medium. The medium was combined and centrifuged at 10,000 g for 30 min to remove cell debris, followed by ultracentrifugation of the supernatants at 100,000 g for 2 h at 4 °C (Beckman coulter, Optima XPN-100). The EV concentration was detected by nanoparticle tracking analysis (NTA, NanoSight NS300) and BCA protein assay kits. The size and potential of EVs were tested with dynamic light scattering (DLS, Zetasizer Nano ZSP). Additionally, the collected medium was incubated with anti-CD63 modified microbeads to capture EVs and then stained with anti-CD9 conjugated Cy5 for flow cytometry analysis (BD Accuri C6).

## Proteomics analysis

The obtained EVs with or without iMASSAGE ($n = 3$/group) were lysed with SDT buffer (4%SDS, 100 mM Tris-HCl, 1 mM DTT, pH 7.6) for protein extraction and then digested with trypsin to obtain peptides. The high pH reversed-phase peptide was detected by liquid chromatography-mass spectrometry on a Q-Exactive spectrometer (Thermo Fisher Scientific). Data were analyzed using MaxQuant

software (1.5.3.17) with false discovery rate (FDR) < 0.01. The reproducibility of identified protein between iMASSAGE and w/o iMASSAGE groups was analysed by Venn diagram. In order to show the significant difference of proteins between the comparison two groups, the fold change (Fold change, FC) of protein expression and $P$ value ($T$ test) were used as the standard to draw a volcano map. The proteins that have been significantly down-regulated were highlighted in blue (FC < 0.5 and $p < 0.05$), while the significantly up-regulated proteins were marked in red (FC > 2.0 and $p < 0.05$), and proteins with no difference were marked as gray.

### Western blotting assay

For western blot analyses, cells or EVs from different treatments of iMASSAGE device were collected and lysed in lysis buffer (20 mM Tris-HCl, pH7.4, 100 mM KCl, 5 mM $MgCl_2$, 0.5% Triton X-100, 1 mM dithiothreitol (DTT), 1 mM phenylmethylsulfonyl fluoride (PMSF) and protease inhibitor cocktail) on ice for 30 min and obtained the supernatant which contains protein. The protein samples were subsequently separated through 12% SDS-PAGE and transferred to nitrocellulose membranes. The blots were treated with 5% non-fat milk or BSA in TBST and left at room temperature for one hour, after which they were incubated with antibodies overnight at a temperature of 4 °C. Following this, they were washed and subjected to a subsequent incubation with HRP-linked secondary antibodies (Cell Signaling Technology). The blots were visualized and analyzed by an enhanced chemiluminescence (ECL) detection system (Tanon 5200).

### The synthesis of AZO-PA hydrogel

4,4′-Di(acrylamido)azobenzene (AZO) was synthesized based on the literature[56]. Then, 3.13 mM AZO was dissolved in mixed solution containing 80 μL dimethyl sulfoxide and 100 μL EtOH. 4.05 M AM were added to 90 μL phosphate-buffered saline (PBS). Finally, AZO and AM solutions were then mixed with APS (10% w/v in water, 10 μL) and 1 μL N,N,N′,N′-tetramethylethylenediamine. AZO-PA hydrogel was formatted in the mold after 30 min of polymerization.

### YAP-mediated EV release

The M2 macrophage-resided iMASSAGE device was cultured in EV-free medium. The iMASSAGE group was treated with cyclic wireless stimulation (1 W, 3 T). The w/o iMASSAGE group was treated without wireless stimulation. The iMASSAGE+VP group was treated with wireless stimulation and pre-incubated with 2 μM verteporfin in the medium. The control (Con) group was cells cultured in commercial culture dishes without any treatment. 24 h later, the hydrogel encapsulated device was cryo-sectioned, fixed with paraformaldehyde (4%) for 15 min and permeabilized with 0.4% Triton X-100 for 8 min. After blocking with 5% BSA solution for 2 h, the slice was incubated with rabbit anti-YAP1 (1:100) for 2 h, followed by incubation with dye-conjugated secondary antibody for 30 min. Finally, slices were observed using confocal microscope. In addition, EVs from all groups in the medium were collected by ultracentrifugation and detected with NTA.

### Intestinal colonization of EVs in vivo

To prove that oral EVs could successfully enter the intestinal lumen for microbiota regulation, we evaluated the stability of EVs in simulated gastric fluid (SGF) and visualized EVs in the intestinal lumen through optical imaging. The iMASSAGE-generated EVs were collected by gradient centrifugation. And then EVs ($4 \times 10^6$ particles/mL) were soaked in SGF for 2 h. The concentration of EVs before and after immersion in SGF was measured by NTA test. Besides, simulated protein (horseradish peroxidase, HRP) and gene (G-quadruplex) were loaded inside EVs by co-incubation. After 2 h of HRP@EV and G-quadruplex@EV immersion in SGF, the activity of HRP and G-quadruplex in EV was detected by kit, respectively. All Animal experiments were carried out

in accordance with ethical guidelines and approved by the Committee for Animal Research of Nanjing University of Posts and Telecommunications (No: 202202). The iMASSAGE-generated EVs ($1 \times 10^{10}$ particles/g) was labeled with Did dye and were orally administered to the female BALB/c mice (20 g). 24 h later, the colon ex vivo was imaged by IVIS spectrum imaging system.

### Assessment of microbiota modulation by iMASSAGE

To construct a mouse model with disordered flora, the female BALB/c mice (20 g) were fed with water containing metronidazole (0.5 g/L), neomycin (1 g/L), vancomycin (1 g/L), and ampicillin (1 g/L) for three days. These mice were randomly divided into three groups (antibiotics, EVs and iMASSAGE). Subsequently, the iMASSAGE-released EVs ($1 \times 10^{10}$ particles/g) were orally administered to mice for three days. On the seventh day, mice feces after different treatments were collected for gut microbiota analysis by 16S rRNA sequencing. Total genomic DNA samples from feces were extracted using the OMEGA Soil DNA Kit (Omega Bio Tek) based on the manufacturer's instructions. Subsequently, PCR amplification of the bacterial 16S rRNA genes V3-V4 region was performed using the forward primer 338 F (5′-ACTCCTACGGGAGGCAGCA-3′) and the reverse primer 806 R (5′-GGACTACHVGGGTWTCTAAT-3′). The 16 S rRNA sequencing was conducted in accordance with the standard protocols of Personabio Bio-pharm Technology Co. Ltd.

### CT imaging and device biocompatibility assessment

The device was surgically implanted into the abdominal cavity of a mouse. Seven days later, mice were imaged by a Computed Tomography (CT, Nano Voxel 1-2702E) to visualize the device location. Additionally, tissue at the device implantation site was analysed by H&E staining.

### Detection of EVs in vivo from device

Abdominal cavity of mice was implanted with device, followed by pulsed wireless stimulation (1 W, 3 T). 24 h later, isolated colon tissue was collected and histologically analysed by CD63 and CD9 immunofluorescence. To further prove that EVs came from cells colonized in device, a stably transfected macrophage (RAW264. 7) line containing the CD63-EGFP-fusion gene was constructed based on previous report[57]. Transfected macrophages were cultured in small dish for 24 h and observed by fluorescence confocal microscopy, whose CD63-GFP protein was detected by western blot. The medium of transfected macrophage was incubated microbeads to capture EVs and then stained with anti-CD9 conjugated Cy5 for flow cytometry analysis. The device colonized with transfected cells was implanted into the intestinal lumen of healthy mice with pulsed wireless stimulation. 24 h later, the obtained colon tissue was imaged for visualization of CD63-GFP protein-containing EVs. Then, homogenized colon tissue and colon contents liquid were incubated with CD9 modified microbeads for EV capture, respectively. These two microbeads were analyzed by flow cytometry.

### Scoring of disease activity index

The clinical disease activity index (DAI) of IBD mice was obtained according to the following three scores: feces consistency (normal, loose, diarrhea), blood stain content in feces (including blood stain content on anal and padding), and weight decline. The scoring of DAI was as described[58] in a blinded fashion.

### In vivo enteritis treatment

To construct an acute colitis model, the female BALB/c mice were continuously fed water containing 2% DSS for five days. Then all mice were divided into three groups ($n = 5$), including PBS, w/o iMASSAGE and iMASSAGE group. On day five, both w/o iMASSAGE and iMASSAGE groups were surgically implanted with devices. The iMASSAGE groups were treated with a wireless impulse signal (1 W, 3 T) from the fifth to

the eighth day. Then, all mice were euthanized to collect biological samples on the tenth day. The weight of the mice was weighed and the state of the feces was observed during the whole process. The cytokines in serum were detected based on the standard protocol of ELISA kits. The MPO activity in colon tissue was assessed with an MPO activity kit (Nanjing Jiancheng, cat. no. A044-1-1). The fixed colon tissue was evaluated by histological analysis (hematoxylin-eosin staining and immunofluorescence).

Fresh mice feces with 50 mg 1-$^{13}$C short-chain fatty acids were added into 1 mL HPLC water containing 10 μg/mL internal standard caproic acid-6, 6, 6-d3. After homogenization, the mixture was kept in an acidic solution (pH = 2–3) for 10 min, followed by centrifugation with 13,000 $g$ for 20 min. Finally, the suspension was stored for further GC-MS analysis. In addition, mice feces with different treatments were collected and stored in a −80 °C refrigerator for gut microbiota analysis by 16 s RNA sequencing.

### ELISA assay
Plasma was gathered by the method of living eyeball blood collection. Then, the collected plasma was centrifuged at 1000 $g$ for a duration of 10 min at room temperature. The upper layer of supernatant, considered serum, was isolated for subsequent cytokine analysis. Tumor necrosis factor-alpha (TNF-α), interleukin-1β (IL-1β), interleukin-6 (IL-6) and interleukin-10 (IL-10) of serum were analyzed by Elisa kits according to vendor's instructions (4 A biotechnology, China).

### Statistics and reproducibility
The data of animal studies were collected in a blinded manner. The experiments have been independently conducted with three times in vitro or five times in vivo for reproducibility of results. General data were analysed by Graphpad prism 8 and origin 2022. No data were excluded from the analyses. Significance was performed with two-tailed Student's $t$ test. *$P < 0.05$, **$P < 0.01$, ***$P < 0.001$, and ****$P < 0.0001$ were considered statistically significant.

### Reporting summary
Further information on research design is available in the Nature Portfolio Reporting Summary linked to this article.

## Data availability
All data of this study are available in the main text and its supplementary files. Any additional requests for information can be directed to, and will be fulfilled by, the corresponding authors. Source data are provided with this paper.

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

## Acknowledgements

We thank Dr. M.W (Suzhou Institute of Nano-Tech and Nano-Bionics, Chinese Academy of Sciences) for the assistance of device establishment and Q.Wu. for her help in animal experiments. This research was supported by National Natural Science Foundation of China (52203170, 22207056, and 62288102), the Natural Science Foundation of Jiangsu Province (BK20220384 and BK20210580) and the Leading-edge Technology Program of Jiangsu Natural Science Foundation (BK20212012) and National University of Singapore Reimagine Grant (A-0009179-02-00 A-0009179-03-00).

## Author contributions

Conceptualization: X.G.D. and D.T.L. Methodology: K.P.W., P.H.H., W.R.L., Z.Y.L., J.C.S., and X.G. Investigation: S.S.W., K.P.W., Y.C.L,, P.H.H,, Z.Y.L,, and X.Z.Z. Visualization: S.S.W., K.P.W., W.J.Y., J.J.Z., and P.H.H. Funding acquisition: L.H.W., S.S.W., X.G.D., and D.T.L. Project administration: S.S.W., X.G.D., and L.H.W. Supervision: S.S.W., X.G.D., D.T.L., and L.H.W. All authors contributed to the writing of the manuscript.

## Competing interests

The authors declare no competing interests.
