## [Peer Review File · Nature Communications]

REVIEWER COMMENTS

Reviewer #1 (Remarks to the Author):

The manuscript entitled "Mechano-electronic stimulation of autologous extracellular vesicle biosynthesis implant for gut microbiota modulation" by ShuangShuang Wan et al. presents the innovative iMASSAGE system. This system simulates in vitro macrophage production of extracellular vesicles (EVs) and demonstrates potential in vivo applications for modulating gut microbiota. Although the technology shows promise, there are several issues that need addressing, particularly regarding its in vivo efficacy. Here are some major concerns:

1. The authors report no significant change in cell numbers after five days culture in the iMASSAGE system, suggesting no cell proliferation despite no toxicity. This raises question about the system's impact on cell proliferation? Additionally, the confocal images of cells provided in the results are too small to discern cell morphology clearly; a magnified view is recommended for better visualization.
2. Proteomic analysis indicates that approximately 20% of proteins change post-iMASSAGE treatment, some markedly so. It would be beneficial to analyze these changed proteins' specific functions and conduct a functional enrichment analysis to determine if iMASSAGE treatment leads to alterations in the abundance of proteins in EVs.
3. The authors administering EVs via gavage and observes significant changes in the gut microbiome. However, given the presence of stomach acid, it's necessary to provide data supporting that the EVs actually reach the intestinal lumen.
4. The paper does not directly cite any studies on the regulatory effects of macrophage M2-derived EVs on the gut microbiome. For instance, the reference in line 295 discusses the therapeutic application of nanopolymers in colitis, which does not directly support the observed phenomena regarding EVs that the authors claimed.
5. While it's generally believed that macrophages produce factors or EVs to regulate gut microbiota, the results suggest that macrophage-derived EVs can promote the growth of gut microbiota post-antibiotic treatment. This hypothesis requires further substantiation through relevant literature, or the authors should discuss these observations more thoroughly.
6. The manuscript claims no significant temperature change with in vivo iMASSAGE implantation, yet Supplementary Fig. 24 shows a difference of at least 2 degrees, which could be statistically significant. This necessitates a more accurate description of the results.
7. Upon in vivo implantation of iMASSAGE, an increase in CD9 and CD63 staining in the intestinal epithelium is observed, predominantly in epithelial cells. Since macrophages are typically sparse in this region, the authors should provide further evidence that the increased EVs are indeed from macrophages, possibly through immunofluorescence colocalization or flow cytometry.

8. Furthermore, evidence is needed to demonstrate that these elevated EVs indeed enter the intestinal lumen and subsequently modulate the gut microbiome.

9. In the DSS model, the authors show an increase in M2 macrophages in the iMASSAGE treatment group. Corresponding evidence is needed to confirm an increase in EVs as well.

Reviewer #2 (Remarks to the Author):

The manuscript describes an innovative technology to induce stiffness changes in hydrogels bearing macrophages to deliver, on demand, therapeutic EVs. Despite interesting and with a very extensive characterization, the reason for the on-demand delivery of EVs for therapeutic purposes is poorly explored; proving that taking advantage of this on-demand responsiveness (stiffness) from the material instead of simply applying the stiffer version of the gel containing cells is necessary. EVs produced after several cycles of contraction/expansion need to be characterized, and their functionality addressed and compared to the one obtained in the first cycles of stimulation. Is the effect of EVs produces in subsequent hydrogel contraction cycles as potent as in the first cycle?

Another control that is required is a hydrogel that will not change its mechanical properties when temperature is varied. A chemically similar hydrogel with similar pore size is recommended for this study. Only this way the effect of the application of radiowaves and subsequent raise in temperature led by the nanoparticles can be isolated from the overall effect of increasing stiffness, leading to the boost in EV production.

Although M2 macrophages were selected as relevant cells to produce therapeutic EVs, the role of this type of stimulation, in continuous or alternated stimulation, needs also to be proved for M1 pro-inflammatory cells, as those will probably be the most common in an inflammatory disease location.

Cells loaded to the device were not characterized sufficiently. How is the 3D distribution of cells? Were they only present at the surface, or throughout the whole device? If cells were neighbouring the nanoparticles, it is unlikely that cycles of heating would not interfere more significantly with cell viability in the gel.

The manuscript would benefit from proving that the concept works in other hydrogels, and providing a maximum limit of stiffness that enables the boosting of EV production.

Reviewer #1:

The manuscript entitled “Mechanoelectronic stimulation of autologous extracellular vesicle biosynthesis implant for gut microbiota modulation” by ShuangShuang Wan et al. presents the innovative iMESSAGE system. This system simulates in vitro macrophage production of extracellular vesicles (EVs) and demonstrates potential in vivo applications for modulating gut microbiota. Although the technology shows promise, there are several issues that need addressing, particularly regarding its in vivo efficacy. Here are some major concerns:

1. The authors report no significant change in cell numbers after five days culture in the iMESSAGE system, suggesting no cell proliferation despite no toxicity. This raises question about the system's impact on cell proliferation? Additionally, the confocal images of cells provided in the results are too small to discern cell morphology clearly; a magnified view is recommended for better visualization.

Reply: Thank you for your concern about the system's impact on cell proliferation. In our study, the hydrogel microenvironment was designed to support cell viability without exerting cytotoxic effects. The stable cell numbers observed over five days suggested an equilibrium where the rate of cell proliferation is balanced by the rate of cell death (Figure 3g). Several factors within the hydrogel environment, such as mechanical properties and composition, potentially influence this balance. For example, studies by Minaisah et al. and Du et al. showed that composition and mechanical cues from hydrogel can significantly influence cell behaviors, including proliferation (*"The use of polyacrylamide hydrogels to study the effects of matrix stiffness on nuclear envelope properties."* *The Nuclear Envelope: Methods and Protocols* (2016): 233-239; *"The correlations between structure, rheology, and cell growth in peptide-based multicomponent hydrogels."* *Polymer Journal* 52.8 (2020): 947-957; *"Hydrogel substrate stress-relaxation regulates the spreading and proliferation of mouse myoblasts."* *Acta biomaterialia* 62 (2017): 82-90.). The variable mechanical properties and composition in our iMESSAGE system influence cellular behavior, potentially limiting proliferation while not causing toxicity.

According to reviewer's suggestion, magnified view of cells had now been added (Figure 3g, inserted).

Fig. 3g, Viability and morphology (upper right) of embedded macrophage cells under multiple days of iMESSAGE treatments with one-day intervals (Calcein AM stained). Scale bar: 10 μ m (up) and 50 μ m (down).

2. Proteomic analysis indicates that approximately 20% of proteins change post-iMESSAGE treatment, some markedly so. It would be beneficial to analyze these changed proteins' specific functions and conduct a functional enrichment analysis to determine if iMESSAGE treatment leads to alterations in the abundance of proteins in

EVs.

Reply: Compared to w/o iMASSAGE groups, iMASSAGE treatment resulted in change in the abundance of 27 protein, of which 17 proteins were down-regulated and 10 proteins were up-regulated (Supplementary Fig. 13b). According to the reviewer's good suggestion, Gene Ontology (GO) enrichment analysis using Fisher's Exact Test was performed to determine changed proteins' specific functions. As shown in Supplementary Fig. 13c, bubble chart exhibited the top 20 GO biological process (BP) terms, indicating these changed proteins were involved in important biological processes such as prostate gland epithelium morphogenesis, prostate gland morphogenesis, Golgi to endosome transport, nitric oxide mediated signal transduction, and prostate gland development. These processes are not associated with inflammatory-involved pathways in macrophages (*Small, 2022, 18(15): 2200060*). Moreover, since most of the proteins (Fig.3e) and key genes that suppress inflammation (Fig.3f) are retained, iMASSAGE-induced EVs show similar function as normally secreted EVs. The new data has been added as Fig. S13c (SI, page 23).

Supplementary Fig. 13c Gene Ontology (GO) enrichment analysis of biological process (BP) using Fisher's Exact Test.

3. The authors administering EVs via gavage and observes significant changes in the gut microbiome. However, given the presence of stomach acid, it's necessary to provide data supporting that the EVs actually reach the intestinal lumen.

Reply: Thank you for your good suggestion. To demonstrate that EVs indeed reached the intestinal lumen, we first evaluated the impact of gastric juice on EVs. We examined the concentration and bioactivity of EVs before and after being dispersed in simulated gastric fluid (SGF). The results demonstrated that EVs could be relatively stable in SGF

in terms of particle concentrations, above 80% of the particle number remained (Supplementary Fig. 22a). To evaluate how the gastric fluid would affect the structure and function of the EVs, we placed the G quadruplex and cholesterol-HRP inside and on the membrane of the EVs respectively as a mimic of their internal nucleic acid enzymes and membrane proteins. The biocatalysis analysis suggested the EVs retained 85% of the bioactivities they carry (Supplementary Fig. 22b). Furthermore, DiD-labeled EVs were orally administered to mice with disturbed gut microbiome. 24 h later, the colons *ex vivo* were imaged. As shown in Supplementary Fig. 22c, there was obvious fluorescence of EVs on the colon compared with control group. These results indicated that EVs could considerably survive with acidic conditions in the stomach and successfully enter the intestine through oral administration. Similar results were observed previously, which may due to the bilayer phospholipid structures of nanovesicles (*Nanomedicine: Nanotechnology, Biology and Medicine* 13.5 (2017): 1627-1636; *Biomaterials* 277 (2021): 121126). The new data has been added as Fig.S27 (SI, page 39).

Supplementary Fig. 27. Intestinal lumen colonization of EVs by oral administration. (a) The concentration of EVs before and after immersion in SGF detected by NTA test. (b) The activity of HRP and G-quadruplex in EV before and after immersion in SGF detected by kit. (c) The iMESSAGE-generated EVs labeled with DiD were imaged after oral administration to healthy mice. Control group (Con) was healthy mice without any treatment. Data present mean \pm s.d. The significant differences were calculated based on a two-tailed Student's t-test. NS refer to no significance. *** $p < 0.001$.

4. The paper does not directly cite any studies on the regulatory effects of macrophage M2-derived EVs on the gut microbiome. For instance, the reference in line 295 discusses the therapeutic application of nanopolymers in colitis, which does not directly support the observed phenomena regarding EVs that the authors claimed.

Reply: Thank you for pointing out this for us. We have revised the references with more relevant studies. While our study propose that macrophage derived-EVs carrying cellular bioactive contents such as miRNA could serve as efficient shuttles to influence microbe through direct or indirect interaction, we acknowledge that direct studies on this specific interaction are limited. The following references provide indirect but relevant support to our results:

(1) EV miRNAs and gut microbiota interaction: We referenced studies that investigated the role of EV miRNAs in colitis, which discusses their interaction with gut microbes. These researches provided insights into the broader landscape of EV-miRNA and their potential regulatory roles in the gut microbiome.

a. Shen, Qichen, et al. *Extracellular vesicle miRNAs promote the intestinal microenvironment by interacting with microbes in colitis.* *Gut Microbes* 14.1 (2022): 2128604.

b. Liu, Shirong, et al. *"The host shapes the gut microbiota via fecal microRNA."* *Cell host & microbe* 19.1 (2016): 32-43.

c. Cai, Qiang, et al. *"Small RNAs and extracellular vesicles: New mechanisms of cross-species communication and innovative tools for disease control."* *PLoS Pathogens* 15.12 (2019): e1008090.

(2) Studies on diverse mediators of macrophage-derived EVs. Additionally, we cited and discussed the diverse roles of macrophage-derived EVs, especially focusing on M2 macrophages. This research sheds light on how M2 macrophages, through EVs, can participate in immunoregulation and tissue repair, potentially impacting gut microbiota. Wang, Yizhuo, et al. *"Macrophage-derived extracellular vesicles: Diverse mediators of pathology and therapeutics in multiple diseases."* *Cell death & disease* 11.10 (2020): 924.

In light of these studies, while direct studies on M2 macrophage-derived EVs and the gut microbiome are scarce, our work contributes to this emerging field by providing new insights and forming a basis for further research. We hope that these amendments and additional references will adequately address the concerns raised and strengthen the manuscript.

5. While it's generally believed that macrophages produce factors or EVs to regulate gut microbiota, the results suggest that macrophage-derived EVs can promote the growth of gut microbiota post-antibiotic treatment. This hypothesis requires further substantiation through relevant literature, or the authors should discuss these observations more thoroughly.

Reply: Thank you for your comment regarding the role of macrophage-derived EVs in gut microbiota regulation. According to your good advice, we expand on this topic and

provide more relevant literature and discussion in the discussion part (Page 10). Recent studies have increasingly recognized the interplay between macrophages and the gut microbiota. Macrophages are known to influence gut microbiota through various mechanisms, including the secretion of cytokines, chemokines, and EVs. EVs are known to carry a myriad of bioactive molecules, including proteins, lipids, and nucleic acids, which can significantly impact the recipient cells and the surrounding microenvironment. For instance, studies by Ji et al. and Liu et al. suggested that host EVs could carry specific miRNA and enable reshape the composition of intestinal gut microbiota (*Biochem Biophys Res Commun* 2018;503:2443-50; *Cell Host Microbe* 2016;19:32–43), suggesting a potential regulatory role of host EVs. EV-miRNA from monocytes and macrophages can further alter Intestinal epithelial cells function and intestinal barrier by modulating the activity of inflammatory transcription genes (*Tissue Barriers* 6.2 (2018): e1431038.).

In our study, we observed an increase in specific bacterial populations in the gut following the administration of M2 macrophage-derived EVs post-antibiotic treatment. This observation aligns with the hypothesis that EVs can promote the growth of gut microbiota. We propose several mechanisms through which this might occur: a) direct stimulation. EVs may directly stimulate the growth of certain bacterial strains by delivering cellular components or substrates (*EMBO reports* 20.3 (2019): e46613). b) immune modulation. By modulating the immune response, EVs could create a more favorable gut environment for the growth of specific bacteria while inhibiting others. We acknowledge that these mechanisms require further experimental validation. Our future research will focus on deciphering the exact components of the EVs that contribute to these effects and the specific bacterial strains that are most responsive.

In conclusion, our findings suggest a significant role of macrophage-derived EVs in gut microbiota regulation. We agree that further molecular studies are needed to fully elucidate these mechanisms.

6. The manuscript claims no significant temperature change with in vivo iMESSAGE implantation, yet Supplementary Fig. 24 shows a difference of at least 2 degrees, which could be statistically significant. This necessitates a more accurate description of the results.

Reply: Thank you for your keen scrutiny. Yes, 2 degrees increase was significantly different. We have revised the description to be around 2.3 °C of temperature increment to be more accurate. To avoid any more misleading comments, we have also performed a detailed statistical analysis in Supplementary Fig. 30. In the manuscript, we have further expanded our discussion to provide a more comprehensive context for the observed temperature change. Additionally, we cited studies that define the thresholds at which temperature variations can significantly affect cellular behavior and tissue integrity (*ACS nano* 9.1 (2015):6-11; *Journal of the American Chemical Society* 136.44 (2014):15684-15693). This enhancement in our discussion supports the notion that the observed temperature increase, despite being statistically noticeable, remains within a range that is physiologically acceptable for the involved tissue and does not exceed levels that could lead to thermal stress or damage.

Supplementary Fig. 30. In vivo photothermal imaging. The images (up) and temperature difference (ΔT , down) of photothermal imaging of implanted devices in mice before and after pulse wireless control. The white circle represents the position of the implanted device. Scale bar: 5 mm. Data present mean \pm s.d. The significant differences were calculated based on a two-tailed Student's t-test. *** $p < 0.001$.

7. Upon in vivo implantation of iMESSAGE, an increase in CD9 and CD63 staining in the intestinal epithelium is observed, predominantly in epithelial cells. Since macrophages are typically sparse in this region, the authors should provide further evidence that the increased EVs are indeed from macrophages, possibly through immunofluorescence colocalization or flow cytometry.

Reply: Thank you for good suggestion, which has prompted us to further substantiate the origin of the increased EVs observed in the intestinal epithelium following in vivo implantation of the iMESSAGE system. To address the concern regarding the source of these EVs, we employed a strategy focusing on the use of a stably transfected RAW264.7 cell line engineered to express a CD63-GFP fusion protein. This approach allowed us to specifically track the EVs produced by the embedded macrophages. The successful creation of the M2 macrophages expressing the CD63-GFP fusion gene was confirmed through the observation of intracellular green fluorescence, as depicted in Supplementary Fig. 32a, alongside a high level of CD63 protein expression shown in Supplementary Fig. 32b. In addition, we also verified that EVs secreted by these transfected M2 macrophages indeed carried the CD63-GFP proteins (Supplementary Fig. 32c). Following this, the stably transfected macrophages were loaded into the iMESSAGE device (Supplementary Fig. 32d).

Upon implantation and subsequent wireless stimulation of the device under specified parameters (1W, 3T), mice were euthanized and their colon tissues were collected for analysis. Flow cytometry analysis of these tissues revealed significant green fluorescence in the colon (Fig. 6c), indicative of the presence of CD63-GFP-tagged EVs. Given EVs derived from endogenous cells within colon tissue did not have green fluorescence emission properties, the detection of EVs exhibiting green fluorescence (32.7%) unambiguously indicates that these EVs originate from the GFP-transfected macrophages loaded in the iMESSAGE system. These new data have been

added as Fig. S32 (SI, Page 44) and Fig 6c (main text, Page23)

Supplementary Fig. 32. Construction of stably transfected macrophages with CD63-GFP gene for EV monitoring in vivo. Transfected M2 macrophages expressing CD63-GFP proteins visualized by (a) confocal microscopy and detected by (b) western blot. Scale bar: 10 μ m. Con refers to normal cell and CD63-GFP refers to M2 macrophages expressing CD63-GFP proteins. (c) Detection of transfected macrophages-produced EV contained CD63-GFP proteins by flow cytometry. (d) The image of iMESSAGE device loaded transfected M2 macrophages and imaged by IVIS imaging system.

Fig.6c Flow cytometry analysis of EVs in colon tissue derived from device embedded stably transduced macrophages with or without pulsed wireless stimulation.

8. Furthermore, evidence is needed to demonstrate that these elevated EVs indeed enter the intestinal lumen and subsequently modulate the gut microbiome.

Reply: Thanks for your good suggestion. To address the concern that the elevated EVs indeed enter the intestinal lumen, we utilized an experimental setup involving the implantation of devices loaded with M2 macrophages transfected with CD63-GFP into mice. Following wireless stimulation, we harvested the colons of these mice for imaging purposes. The imaging results, presented in Supplementary Fig. 34a, clearly demonstrate the presence of GFP fluorescence within the colon tissues. This fluorescence is attributable to CD63-GFP-tagged EVs secreted by the iMESSAGE system, providing direct evidence of EV entry into the intestinal lumen. To further validate this observation, we collected luminal contents for flow cytometry analysis. The analysis revealed a markedly stronger fluorescence signal in the iMESSAGE-

treated group compared to the group without iMASSAGE treatment, as shown in Supplementary Fig. 34b. This indicates a higher accumulation of CD63-GFP-EVs in the intestinal lumen of the treated group. Collectively, the results presented in Figure 5, Figure 6d-f and Supplementary Fig. 35 suggested that M2-derived EVs could indeed enter the intestinal lumen for the modulation of disordered gut microorganisms. The new data has been added as Fig. S34 (SI, Page 46)

Supplementary Fig. 34. Detection of EVs produced by iMASSAGE device with stably transduced macrophage colonization in the intestinal lumen. (a) Fluorescent images of ex vivo colon in the group treated with pulsed wireless (1W, 3T). Ex: 488 nm, Em: 509 nm. (b) Flow cytometry analysis of EV containing CD63-GFP protein concentration in intestinal lumen contents after pulsed wireless stimulation (1W, 3T).

9. In the DSS model, the authors show an increase in M2 macrophages in the iMASSAGE treatment group. Corresponding evidence is needed to confirm an increase in EVs as well.

Reply: Thanks for your good suggestion. In Supplementary Fig. 33, we utilized staining for EV biomarkers, specifically CD63 and CD9, to illustrate the augmentation of EVs within colon tissue following iMASSAGE intervention. This staining provided visual confirmation of the increased presence of EVs, which was consistent with the enhancement in M2 macrophages. To build on this evidence and specifically demonstrate the targeted increase of EVs in the intestinal lumen, we employed stably transfected M2 macrophages within the iMASSAGE system. This strategic approach allowed us to trace and quantify the EVs originating from these macrophages. The results, as depicted in Supplementary Fig. 34, clearly indicated an exclusive increase in EVs within the intestinal lumen content following iMASSAGE treatment.

Supplementary Fig. 34. Detection of EVs produced by iMASSAGE device with stably transduced macrophage colonization in the intestinal lumen. (a) Fluorescent images of ex vivo colons in the group treated with pulsed wireless (1W, 3T). Ex: 488 nm, Em: 509 nm. (b) Flow cytometry analysis of EV containing CD63-GFP protein concentration in intestinal lumen contents after pulsed wireless stimulation (1W, 3T).

Reviewer #2:

1. The manuscript describes an innovative technology to induce stiffness changes in hydrogels bearing macrophages to deliver, on demand, therapeutic EVs. Despite interesting and with a very extensive characterization, the reason for the on-demand delivery of EVs for therapeutic purposes is poorly explored; proving that taking advantage of this on-demand responsiveness (stiffness) from the material instead of simply applying the stiffer version of the gel containing cells is necessary.

Reply: Thank you for your constructive comments to our work. We recognize the importance of clearly articulating the rationale behind the on-demand stiffness modulation of our hydrogel system for the production of therapeutic EVs. The essence of our innovation lies not just in the ability to alter hydrogel stiffness but in harnessing this property to provide a tailored therapeutic response. Here, we elaborate on the following considerations to better justify the necessity and advantages of our approach:

(1) Adaptability to disease progression: Diseases often exhibit dynamic progression, with phases of exacerbation and remission. A static hydrogel, regardless of its stiffness, cannot accommodate the variable therapeutic demands dictated by the changing severity of disease states. Our on-demand system, however, can modulate the release of EVs in response to the specific stage of disease or healing process, allowing for a more tailored therapeutic approach that matches the temporal dynamics of the disease.

(2) Controlled dosing and minimized side effects: Continuous exposure to therapeutic agents, such as EVs, can lead to therapeutic resistance or adverse side effects due to overstimulation of target tissues. An on-demand system mitigates these risks by precisely controlling the dosage and timing of EV release, ensuring that therapy is administered only when beneficial, thus preserving therapeutic efficacy and minimizing potential side effects.

(3) Enhanced therapeutic outcomes through precision medicine: The ability to adjust stiffness and, consequently, EV release in real-time allows for personalized treatment strategies. This level of control ensures that patients receive the optimal dosage at the right time, potentially leading to better therapeutic outcomes compared to a one-size-fits-all approach.

(4) Integration with external control systems: The dynamic nature of the on-demand hydrogel system facilitates integration with external monitoring and control systems. This could allow for real-time adjustments based on physiological feedback, further enhancing the precision and adaptability of the therapy.

In the revision, this clarification has been added in the discussion part to justify the necessity and the advantages of on-demand delivery of EV system (Main text, Page 9).

2. EVs produced after several cycles of contraction/expansion need to be characterized, and their functionality addressed and compared to the one obtained in the first cycles of stimulation. Is the effect of EVs produces in subsequent hydrogel contraction cycles as potent as in the first cycle?

Reply: Thank you for the good suggestion. In the revision, we specifically focused on EVs generated after the third cycle of stimulation, selected based on optimized

parameters that ensure cell viability. Quantitative analysis indicated that the production concentration of EVs in the third cycle remained robust, as detailed in Supplementary Fig. 18a. Importantly, our investigations into the morphology of EVs across different stimulation cycles revealed no significant changes, ensuring the physical integrity of EVs is maintained over repeated cycles (Supplementary Fig. 18b). Furthermore, the analysis of proteins and genetic materials within the EVs demonstrated that most proteins (Supplementary Fig. 18c) and important genes (Supplementary Fig. 18d) are preserved across cycles. The new data has been added as Fig. S18 (SI, Page 30).

Supplementary Fig. 18. Differences in EVs produced by iMASSAGE device between single- and multiple-pulse wireless stimulation. (a) Concentration of EVs released from iMASSAGE with first stimulation and third stimulation. (b) TEM image of iMASSAGE-released EVs by the third stimulation. (c) SDS-PAGE protein analysis of EV from iMASSAGE with first cycle stimulation and third cycle stimulation. (d) The relative expression of miRNA-125b and miR-149 from iMASSAGE-released EVs with first stimulation and third stimulation. The significant differences were calculated based on a two-tailed Student's t-test. ** $p < 0.01$, ns refers to no significant.

To assess the *in vivo* biological efficacy of EVs produced at various stimulation intervals, we examined their effect on the concentration of short-chain fatty acids (SCFAs) in the colon of mice with a disrupted gut microbiome. This measure serves as an indicator of the therapeutic potential of M2-derived EVs in modulating gut microbiota. Our findings, as illustrated in Figure 1, show that oral administration of EVs derived from both the first and third cycles of stimulation (administered in equal quantities), led to a comparable increase in SCFA levels in the gut. This outcome suggests that despite the EVs produced across different stimulation cycles, the functional components and, consequently, the therapeutic efficacy of the EVs remain consistent.

Supplementary Fig. Comparative analysis of acetic and butyric acid concentrations in mice feces following treatment with EVs produced by iMESSAGE device under first cycle and third cycle stimulation.

3. Another control that is required is a hydrogel that will not change its mechanical properties when temperature is varied. A chemically similar hydrogel with similar pore size is recommended for this study. Only this way the effect of the application of radiowaves and subsequent raise in temperature led by the nanoparticles can be isolated from the overall effect of increasing stiffness, leading to the boost in EV production.

Reply: Thank you for the good suggestion. To verify that the iMESSAGE-induced EV produce was mainly regulated by the effect of variable stiffness, we incorporated a chemically similar polyacrylamide (PAM) hydrogel as a control device within our study. This PAM hydrogel, referenced from *Biomaterials* 27.35 (2006): 5883-5891, was known for its non-thermo-responsive mechanical properties, making it an ideal candidate to exclude the potential confounding effects of exogenous radiowave stimulation and temperature changes. To ensure a fair comparison, the PAM hydrogel control device was assembled using a process akin to that of the iMESSAGE device, including the encapsulation of wirelessly powered μ LEDs and photothermal cuttlefish ink nanoparticles during the hydrogel synthesis. This approach allowed for the activation of the PAM device through wireless stimulation, as detailed in Supplementary Fig. 14a. Importantly, the PAM device was engineered to exhibit a pore size similar to that of the iMESSAGE device, approximately 50 μ m (Supplementary Fig. 14b), ensuring consistency in physical structure between the control and experimental setups. Upon evaluating the mechanical behavior of the PAM device under conditions of wireless stimulation (2 minutes On, 6 minutes Off), we observed no significant changes in volume, pore size, or Young's modulus, confirming the hydrogel's stable mechanical properties under these conditions (Supplementary Fig. 14c-e). This stability is crucial for our control experiments. Furthermore, we conducted a comparative analysis of EV production from the PAM device under both stimulated and non-stimulated conditions. Utilizing Nanoparticle Tracking Analysis (NTA) and Bicinchoninic Acid (BCA) assay results (Supplementary Fig. 14f,g), we found no significant difference in EV production between the two conditions. This outcome

strongly supports the conclusion that the increase in EV production observed with the iMASSAGE device can indeed be attributed to the variable mechanical properties induced by wireless stimulation, rather than to thermal effects or the presence of exogenous radiowaves. This control experiment, therefore, effectively isolates the impact of the iMASSAGE device's variable stiffness on EV production, affirming the specificity of our findings to the mechanical changes.

The new data has been added as Fig. S14 (SI, Page 25).

Supplementary Fig. 14. Construction and performance evaluation of PAM device. (a) The image of PAM device activated by wireless. (b) Representative SEM image of pore size of PAM device. Scale bar: 50 μm . (c) The temperature change of the device over wireless time. (d) Representative SEM images of the internal aperture of PAM device in its original state (0 min), the wireless-induced contraction state (2 min) and relaxation state (8 min). Scale bar: 50 μm . (e) The relative volume and (f) Young's modulus of

PAM device at 0, 2 and 8 min. PAM-produced EV concentration detected by NTA (g) and (h) BCA kit with or without stimulation. The significant differences were calculated based on a two-tailed Student's t-test. ns refers to no significance.

4. Although M2 macrophages were selected as relevant cells to produce therapeutic EVs, the role of this type of stimulation, in continuous or alternated stimulation, needs also to be proved for M1 pro-inflammatory cells, as those will probably be the most common in an inflammatory disease location.

Reply: Thank you for your suggestion. Following the suggestion, M1 macrophages were loaded into iMESSAGE device and the concentration of produced EVs were examined by the BCA kit under the condition of pulsed wireless stimulation. The result suggested EV protein concentration of the M1-resided device under pulse stimulation was 11 times higher than that without stimulation (Supplementary Fig. 23). This observation underscores the ability of the iMESSAGE device to induce EV secretion from M1 macrophages. To further validate that the increased EV secretion was specifically a result of stimulation applied to cells within the device rather than an external influence from macrophages not incorporated within the device, we conducted a control experiment by implanting cell-free devices in vivo. The specific vesicle CD63 staining in tissues revealed no significant difference between the wireless-stimulated and non-stimulated groups (Supplementary Fig. 31). These outcomes confirm that the augmented secretion of EVs is indeed a direct consequence of activating the mechanosensitive cells housed within the iMESSAGE device.

Supplementary Fig. 23. Relative concentrations of EVs produced under iMESSAGE or not with M1 macrophage embedded. The significant differences were calculated based on a two-tailed Student's t-test. ***p < 0.001.

Supplementary Fig. 31. Specific anti-CD63 staining of tissues near the implant without cell loading. Scale bar: 20 μ m.

5. Cells loaded to the device were not characterized sufficiently. How is the 3D distribution of cells? Were they only present at the surface, or throughout the whole device? If cells were neighbouring the nanoparticles, it is unlikely that cycles of heating would not interfere more significantly with cell viability in the gel.

Reply: Thank you for your comment. To address your concern regarding the characterization of cell distribution within the device, we utilized confocal imaging to analyze the three-dimensional (3D) distribution of macrophages post-incubation. Twelve hours after incubation with the device, confocal images revealed that the macrophages, marked by green fluorescence, were dispersed randomly throughout the hydrogel matrix. This distribution pattern can be attributed to the hydrogel's large pore size, which facilitates cell penetration and distribution, and the presence of RGD motifs within the hydrogel, which promote cell adhesion and entrapment (Supplementary Fig. 10a).

Furthermore, the 3D imaging provided insights into the spatial relationship between the cells and the photothermal particles (CINPs, red labeled) embedded within the device. The analysis showed that the proximity of cells to photothermal particles varied randomly, with a generally low incidence of direct adjacency. This observation is particularly relevant in the context of the concentration of photothermal particles used in our study, which was optimized to minimize potential adverse effects on cell viability. To directly assess the impact of the wireless-thermal effect generated by the device on cell viability, we conducted comparative analyses using lactate dehydrogenase (LDH)

release in both stimulated and non-stimulated conditions. The result, as presented in Supplementary Fig. 10b, demonstrated that cell viability within the device remained high and comparable to that of the control group, even after wireless stimulation. This finding indicates that the periodic heating generated by the device did not significantly affect the overall viability of the colonized cells.

The new data has been added as Fig. S10 (SI, Page 20).

Supplementary Fig. 10. (a) Representative 3D confocal image of iMASSAGE hydrogel showing the distribution of macrophages (green labeled) and CINPs (red labeled). Scale bar: 50 μm. (b) LDH release of cells embedded in the device with/without pulsed wireless stimulation. The significant differences were calculated based on a two-tailed Student's t-test. ns refer to no significant.

6. The manuscript would benefit from proving that the concept works in other hydrogels, and providing a maximum limit of stiffness that enables the boosting of EV production.

Reply: Thank you for your good suggestion. We explored the applicability of our approach using a photo-responsive 4,4'-di(acrylamido) azobenzene cross-linked polyacrylamide (AZO-PA) hydrogel, which exhibits variable stiffness in response to external light stimuli (*ACS applied materials & interfaces* 10.9 (2018): 7765-7776.). The synthesized AZO-PA hydrogel demonstrated reversible mechanical changes, with stiffness increasing from 2.3 KPa to 10.5 KPa under 490 nm light exposure for 10 minutes and subsequently reducing to 5.1 KPa with 365 nm light irradiation for 5 minutes (Supplementary Fig. 24a). Embedding M2 macrophages into this hydrogel, we observed a photo-responsive increase in EV release, with the concentration of EVs from the cell-embedded hydrogel under light stimulation being 4.4 times higher than without stimulation (Supplementary Fig. 24b). This result supports the versatility of our concept across different hydrogel systems.

Regarding the stiffness threshold for EV production, our results revealed a direct correlation between the increase in Young's modulus of the iMASSAGE device and the wireless input power, with EV concentration also showing a positive relationship with

input power (Supplementary Fig. 7c,d and Supplementary Fig. 15). At the highest tested wireless input power of 2W, the device achieved a Young's modulus of 32.3 kPa, correlating with the peak in EV production (approximately 11.3×10^7 particles/mL EVs). However, cell viability assessments at varying input powers indicated that a 2W setting could potentially compromise cell integrity due to exceeding the cells' stiffness tolerance (Supplementary Fig. 16). Therefore, based on our findings, we identified 1W as the optimal wireless input power for our experiments, striking a balance between maximizing EV production and preserving cell viability within the device. This setting corresponds to a maximum effective stiffness of 32.3 kPa, which not only supports enhanced EV release but also ensures the longevity and functional integrity of the device.

The new data has been added as Fig. S24 (SI, Page 36) and Fig. S16 (SI, Page 28).

Supplementary Fig. 24. (a) Young's modulus of AZO-PA hydrogel by irradiation. AZO-PA hydrogel was exposed to UV light (365 nm) for 10 min, blue light (490 nm) for 2 h and then UV light (365 nm) for 10 min. (b) The concentration of EVs by NTA with or without stimulation (photo stimulation: 10 min of UV, 2 h of blue and 10 min of UV). Data present mean \pm s.d. The significant differences were calculated based on a two-tailed Student's t-test. **** $p < 0.001$.

Supplementary Fig. 16. The viability of colonized cells in iMESSAGE treated with different input power by LDH release. The significant differences were calculated based on a two-tailed Student's t-test. **p < 0.01, ns refers to no significant.

REVIEWERS' COMMENTS

Reviewer #1 (Remarks to the Author):

The author has addressed my concerns.

Reviewer #2 (Remarks to the Author):

The authors addressed all concerns in a satisfactory manner. Therefore, I recommend publication.